# A model for learning based on the joint estimation of stochasticity and volatility

Payam Piray [1✉] & Nathaniel D. Daw [1]

Previous research has stressed the importance of uncertainty for controlling the speed of learning, and how such control depends on the learner inferring the noise properties of the environment, especially volatility: the speed of change. However, learning rates are jointly determined by the comparison between volatility and a second factor, moment-to-moment stochasticity. Yet much previous research has focused on simplified cases corresponding to estimation of either factor alone. Here, we introduce a learning model, in which both factors are learned simultaneously from experience, and use the model to simulate human and animal data across many seemingly disparate neuroscientific and behavioral phenomena. By considering the full problem of joint estimation, we highlight a set of previously unappreciated issues, arising from the mutual interdependence of inference about volatility and stochasticity. This interdependence complicates and enriches the interpretation of previous results, such as pathological learning in individuals with anxiety and following amygdala damage.

[1] Princeton Neuroscience Institute and Department of Psychology, Princeton University, Princeton, NJ, USA. ✉email: ppiray@princeton.edu

Among the successes of computational neuroscience is a level-spanning account of learning and conditioning, which has grounded biological plasticity mechanisms (specifically, error-driven updating) in terms of a normative analysis of the problem faced by the organism[1–5]. These models recast learning as statistical inference: using experience to estimate the amount of some outcome (e.g., food) expected on average following some cue or action. This is an important sub-problem of reinforcement learning, which uses such value estimates to guide choice. The statistical framing has motivated an influential program of investigating the brain's mechanisms for tracking uncertainty about its beliefs, and how these impact learning[6–10].

Uncertainty, in turn, depends upon the noise properties of the values being learned, including both the degree of stochasticity in their measurement (observation noise, the variance of which we call *stochasticity*) and how quickly or how often they change (process noise, the variance of which is known as *volatility*). In general, then, a statistically efficient learner must estimate not only just the primary quantity of interest (e.g., the action value) but also parameters describing its noise properties. This perspective has inspired a series of hierarchical Bayesian inference models, which extend inference to either volatility or stochasticity, though typically while treating the other noise parameter as fixed and known to the model.

For volatility, a particularly influential series of theories extends a baseline model known as the Kalman filter to incorporate volatility estimation (but conditional on known stochasticity)[6,11,12]. All else equal, when volatility is higher, the organism is more uncertain about the cue's value (because the true value will on average have fluctuated more following each observation), and so the learning rate (the reliance on each new outcome) should be higher. A series of experiments have reported behavioral and neural signatures of these effects of volatility enhancing learning rate, and also their disruption in relation to psychiatric symptoms[6,8,10,13–23]. Conversely, stochasticity also affects the learning rate, but in the opposite direction: all else equal, when individual outcomes are more stochastic (larger stochasticity), they are *less* informative about the cue's true value and the learning rate, in turn, should be *smaller*. Here again, experiments confirm that people adjust their learning rates in the predicted direction[24,25], and this behavior has been captured by a model that estimates stochasticity (but treating the process noise parameter, in this case a hazard rate, as known).

Altogether, this work has led to a strong argument that the brain's mechanisms for tracking uncertainty, and the inference of the noise parameters that govern it, are crucial to healthy and disordered learning[26]. Although these components seem individually well understood, in this article we argue that important insights are revealed by considering in greater detail the full problem facing the learner: simultaneously estimating both volatility and stochasticity during learning. By introducing a model that performs such joint estimation, and studying its behavior in reinforcement learning tasks, we show that because of the interrelationship between these variables, a full account of any of them interacts with the other in consequential ways.

The key issue is that although the learner's estimates of them play opposite roles on learning rates, from the learner's perspective, they both similarly manifest in noisier, less reliably predictable outcomes. The observation that experienced noise can, to a first approximation, be explained by either volatility or stochasticity—and that these effects might be confused, either by experimenters or by learners—has implications. First, previous work apparently showing variation in volatility processing in different groups, such as various psychiatric patients[14,16,17,19,21,22,27,28] (using a model and tasks that do not vary stochasticity), might instead reflect misidentified abnormalities in processing stochasticity. We suggest that future research should test both the dimensions of learning explicitly. Furthermore, from the perspective of a learner inferring volatility and stochasticity, these factors should compete or trade off against one another to best explain experienced noise. This means that any dysfunction or damage that impairs detection of stochasticity, should lead to a compensatory increase in inferred volatility, and vice versa: a classic pattern known in Bayesian inference as a failure of "explaining away."

We argue that such compensatory tradeoffs may be apparent both in anxiety disorders and following damage to amygdala. Intolerance of uncertainty is thought to be a critical component of anxiety and a crucial risk factor for developing anxiety disorders[29,30]. Although there has been recent interest in operationalizing this idea by connecting it to statistical learning models and tasks[26,31–38], we and others have focused on apparent abnormalities in processing volatility[13,20,26]. The current model suggests a different interpretation, in which anxiety primarily disrupts inference about stochasticity, but with the additional result that the learner misinterprets noise due to stochasticity as a signal of change, i.e., volatility. We argue that the complementary pattern of explaining away, in which a failure to detect volatility leads to change misattributed to stochasticity, can be appreciated in studies of the amygdala's role in modulating learning rates. In particular, our model suggests that a specific involvement of amygdala in volatility (and the explaining away pattern) explains effects of amygdala damage better than an involvement in learning rates more generally. These sorts of reciprocal interactions also give rise to a richer and subtler set of possible patterns of dysfunction that may help to understand a wide range of other neurological and psychiatric disorders, such as schizophrenia, in which there has been a tendency to study altered processing of uncertainty narrowly in the context of volatility.

The model also sheds light on experimental phenomena of learning rates in conditioning, and on two classic descriptive theories of conditioning in psychology that have been interpreted as predecessors of the statistical accounts[39,40]. We suggest that seemingly contradictory effects of noise in different experiments, associated with these two theories, can be identified with effects on learning rates of inferred stochasticity vs. volatility, respectively. On this view, different effects will dominate in different experiments depending on which parameter the pattern of the noise suggests.

In the remainder of this article, we present i) a probabilistic model for the joint estimation of volatility and stochasticity from experience; and ii) volatility- and stochasticity-lesioned models in which the corresponding module is damaged. These models highlight the mutual interdependence of inference about volatility and stochasticity and show how the interdependence leads the model to predict paradoxical compensatory behaviors if inference about either factor is damaged. We use these lesioned models in a series of simulation experiments to explain aspects of pathological behavior observed in anxiety disorders and following amygdala damage.

## Results

**Model**. We begin with the Kalman filter, which describes statistically optimal learning from data produced according to a specific form of noisy generation process. The model assumes that the agent must draw inferences (e.g., about true reward rates) from observations (individual reward amounts) that are corrupted by two distinct sources of noise: process noise or *volatility* and outcome noise or *stochasticity* (Fig. 1a, b). Volatility captures the speed by which the true value being estimated changes from

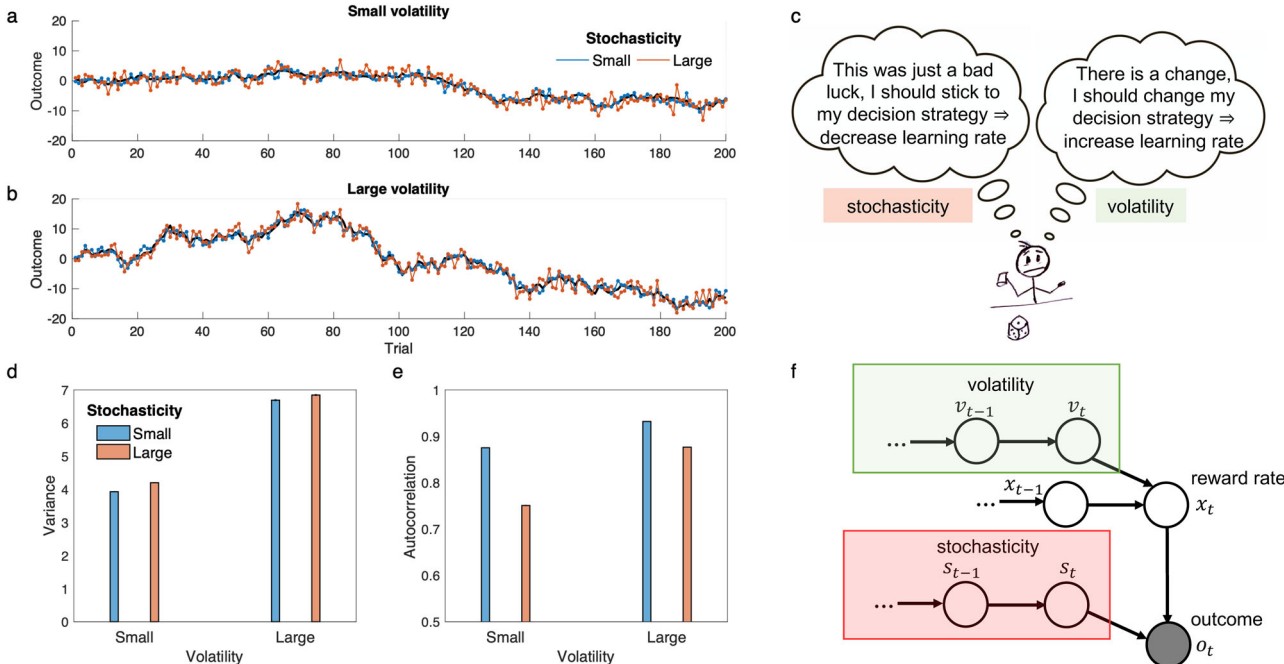

**Fig. 1 Statistical difference between volatility and stochasticity. a–b** Examples of generated time-series based on a small and large constant volatility parameter given a small (**a**) or a large (**b**) constant stochasticity parameter are plotted. **c** Given a surprising observation (e.g., a negative outcome), one should compute how likely the outcome is due to the stochasticity (left balloon) or due to the volatility (right balloon). Dissociating these two terms is important for learning, because they have opposite influences on learning rate. **d–e** It is possible to infer both volatility and stochasticity based on observed outcomes, because these parameters have dissociable statistical signatures. In particular, although both of them increase variance (**d**), but they have opposite effects on autocorrelation (**e**). In particular, whereas volatility increases autocorrelation, stochasticity tends to reduce it. Here, 1-step autocorrelation (i.e., correlation between trial $t$ and $t-1$) was computed for 100 time-series generated with parameters defined in **b** and **c**. Small and large parameters for volatility were 0.5 and 1.5 and for stochasticity were 1 and 3, respectively. **f** Structure of the (generative) model: outcomes were stochastically generated based on a probabilistic model depending on reward rate, stochasticity and volatility. Only outcomes were observable (the gray circle), and value of all other parameters should be inferred based on outcomes. The observed outcome is given by the true reward rate, $x_t$, plus some noise whose variance is given by the stochasticity, $s_t$. The reward rate itself depends on its value on the previous trial plus some noise whose variance is given by the volatility, $v_t$. Both volatility and stochasticity are dynamic and probabilistic Markovian variables generated noisily based on their value on the previous trial. Thus, the model has two modules, volatility and stochasticity, which compete to explain experienced noise in outcomes. See Methods for formal treatment of the model. Errorbars in (**d–e**) are standard error of the mean calculated across 10000 simulations and are too small to be visible. Source data are provided as Source Data file.

trial to trial (modeled as Gaussian diffusion); stochasticity describes additional measurement noise in the observation of each outcome around its true value (modeled as Gaussian noise on each trial).

For this data generating process, if the true values of volatility and stochasticity, $v_t$ and $s_t$ are known, then optimal inference about the underlying reward rate is tractable using a specific application of Bayes rule, here called the Kalman filter[41]. The Kalman filter represents its beliefs about the reward rate at each step as a Gaussian distribution with a mean, $m_t$, and variance (i.e., uncertainty about the true value), $w_t$. The update, on every trial, is driven by a prediction error signal, $\delta_t$, and learning rate, $\alpha_t$. This leads to simple update rules following observation of outcome $o_t$:

$$\delta_t = o_t - m_t \tag{1}$$

$$\alpha_t = \frac{w_t + v_t}{w_t + v_t + s_t} \tag{2}$$

$$m_{t+1} = m_t + \alpha_t \delta_t \tag{3}$$

$$w_{t+1} = (1 - \alpha_t)(w_t + v_t) \tag{4}$$

This derivation thus provides a rationale for the error-driven update (Eq. 3) prominent in neuroscience and psychology[42], and adds to these a principled account of the learning rate, $\alpha_t$, which

on this view should depend (Eq. 2) on the agent's uncertainty and the noise characteristics of the environment. In particular, Eq. 2 shows that the learning rate is increasing and decreasing, respectively, with volatility and stochasticity. This is because higher volatility increases the chance that the true value will have changed since last observed (increasing the need to rely on the new observation), but higher stochasticity decreases the informativeness of the new observation relative to previous beliefs.

This observation launched a line of research focused on elucidating and testing the prediction that organisms adopt higher learning rates when volatility is higher[6]. But the very premise of these experiments violates the simplifying assumption of the Kalman filter—that volatility is fixed and known to the agent. To handle this situation, new models were developed[11,12] that generalize the Kalman filter to incorporate learning the volatility $v_t$ as well, arising from Bayesian inference in a hierarchical generative model in which the true $v_t$ is also changing. In this case, exact inference is no longer tractable, but approximate inference is possible and typically incorporates Eqs. 1–4 as a subprocess.

This line of work on volatility estimation inherited from the Kalman filter the view of stochasticity as fixed and known. Of course, in general, all the same considerations apply to stochasticity as well: it must be learned from experience, may be changing, and its value impacts learning rate. Indeed,

estimating both parameters is critical for efficient learning, because they have opposite effects on learning rate: whereas volatility increases the learning rate, stochasticity reduces it (Fig. 1c). Although some algorithms have been explored for learning when both noise parameters are unknown[43–45], the main other application of this type of model in neuroscience has relied on a different simplification[25], which estimates the stochasticity while treating the hazard rate (the analogue of volatility for changepoint problems) as fixed and known to the model.

Learning these two parameters simultaneously is more difficult because, from the perspective of the agent, larger values of either volatility or stochasticity result in more surprising observations: i.e., larger outcome variance (Fig. 1e). However, there is a subtle and critical difference between the effects of these parameters on generated outcomes, whereas larger volatility increases the autocorrelation between outcomes (i.e., covariation between the outcomes on consecutive trials), stochasticity reduces the autocorrelation (Fig. 1f). This is the key point that makes it possible to dissociate and infer these two terms while only observing outcomes.

We developed a probabilistic model for learning under these circumstances. The data generation process arises from a further hierarchical generalization of these models (specifically the generative model used in our recent work[12]), in which the true value of stochasticity is unknown and changing, as are the true reward rate and volatility (Fig. 1d). The goal of the learner is to estimate the true reward rate from observations, which necessitates inferring volatility and stochasticity as well.

As with models of volatility, exact inference for this generative process is intractable. Furthermore, in our experience this problem is also relatively challenging to handle with variational inference, the family of approximate inference techniques used previously (see Discussion). Thus, we have instead used a different standard approximation approach that has also been popular in psychology and neuroscience, Monte Carlo sampling[3,46–48]. In particular, we use particle filtering to track $v_t$ and $s_t$ based on data[49,50]. Our method exploits the fact that given a sample of volatility and stochasticity, inference for the reward rate is tractable and is given by Eqs. 1–4, in which $s_t$ and $v_t$ are replaced by their corresponding samples (see Methods; this combination of sequential sampling with exact inference for a subproblem is known as Rao-Blackwellized particle filtering).

**Learning under volatility and stochasticity**. We now consider the implications of this model for learning under volatility and stochasticity.

A series of studies has used two-level manipulations (high vs. low volatility blockwise) to investigate the prediction that learning rates should increase under high volatility[6,13,38,51]. Here volatility has been operationalized by frequent or infrequent reversals (Fig. 2a), rather than the smoother Gaussian diffusion that the volatility-augmented Kalman filter models formally assume. Nevertheless, applied to this type of task, these models detect higher volatility in the frequent-reversal blocks, and increase their learning rates accordingly[6,11,12]. The current model (which effectively incorporates the others as a special case) achieves the same blockwise result when stochasticity is held fixed across both blocks (Supplementary Fig. 1).

In the preceding line of studies, stochasticity was not manipulated. (Indeed, it was not even independently manipulable because rewards were binary, and the variance of binomial outcomes is determined only by the mean.) However, analogous effects of stochasticity have been seen in another line of studies[7,9,25,52]. In these studies, Nassar and colleagues studied learning rates in a task in which subjects had to predict a value,

from observations in which the true value was corrupted, blockwise, by different levels of additive Gaussian noise (i.e., stochasticity) and occasionally "jumping" with a constant hazard rate, analogous to volatility. The main feature of these results relevant to the current model is that these studies have shown that participants' learning rate decreases with increases in the noise level (see also[24]). This effect cannot be explained by models that only consider volatility, and in fact, those models make opposite predictions because they take increased noise as evidence of volatility increase. The current model, however, produces the same blockwise effect as humans: because it correctly infers the change in stochasticity, its learning rate is lower, on average, for higher levels of noise (Supplementary Fig. 2). Although we do not intend the current model as a detailed account of how people solve this class of tasks (which is based on a somewhat different generative dynamics), the model can also reproduce other more fine-grained aspects of human behavior in this task, particularly increases in learning rate following switches and scaling of learning rate with the magnitude of error (Supplementary Fig. 2).

Note that while considered together, these two lines of studies separately demonstrate the two types of effects on learning rates we stress, neither of these lines of work has manipulated stochasticity alongside volatility (though see also[24]). Furthermore, learning of the noise hyperparameters in these studies has largely been explicitly modeled only for either parameter conditional on the other being known. We next consider a variant of this type of task, elaborated to include a 2 × 2 factorial manipulation of both the stochasticity alongside volatility (Fig. 2; we also substitute smooth diffusion for reversals). Here, both parameters are constant within the task, but they are unknown to the model. A series of outcomes was generated based on a Markov random walk in which the hidden reward rate is changing according to a random walk and the learner observes outcomes that are noisily generated according to the reward rate.

Figure 2 shows the model's learning rates and how these follow from its inferences of volatility and stochasticity. As above, the model increases its learning rate in the higher volatility conditions but as expected it also decreases it in the higher stochasticity conditions (Fig. 2a). These effects on learning rate arise, in turn (via Eq. 2) because the model is able to correctly estimate the various combinations of volatility and stochasticity from the data (Fig. 2b, c).

Our model thus suggests a general program of augmenting the standard 2-level volatility manipulation by crossing it with a second manipulation, of stochasticity, and predicts that higher stochasticity should decrease learning rate, separate from volatility effects.

**Interactions between volatility and stochasticity**. The previous results highlight an important implication of the current model: that inferences about volatility and stochasticity are mutually interdependent. These interrelationships immediately imply a general interpretational issue for experiments that manipulate only one of these noise parameters, and analyze data using a model that attributes all dynamic learning rate effects to one of them. But the details of the interdependence are themselves informative. From the learner's perspective, a challenging problem (simplified away in many of the previous models) is to distinguish volatility from stochasticity when both are unknown, because both of them increase the noisiness of observations. Disentangling their respective contributions requires trading off two opposing explanations for the pattern of observations, a process known in Bayesian probability theory as *explaining away*. Thus, models that neglect stochasticity tend to misidentify stochasticity as volatility and inappropriately modulate learning.

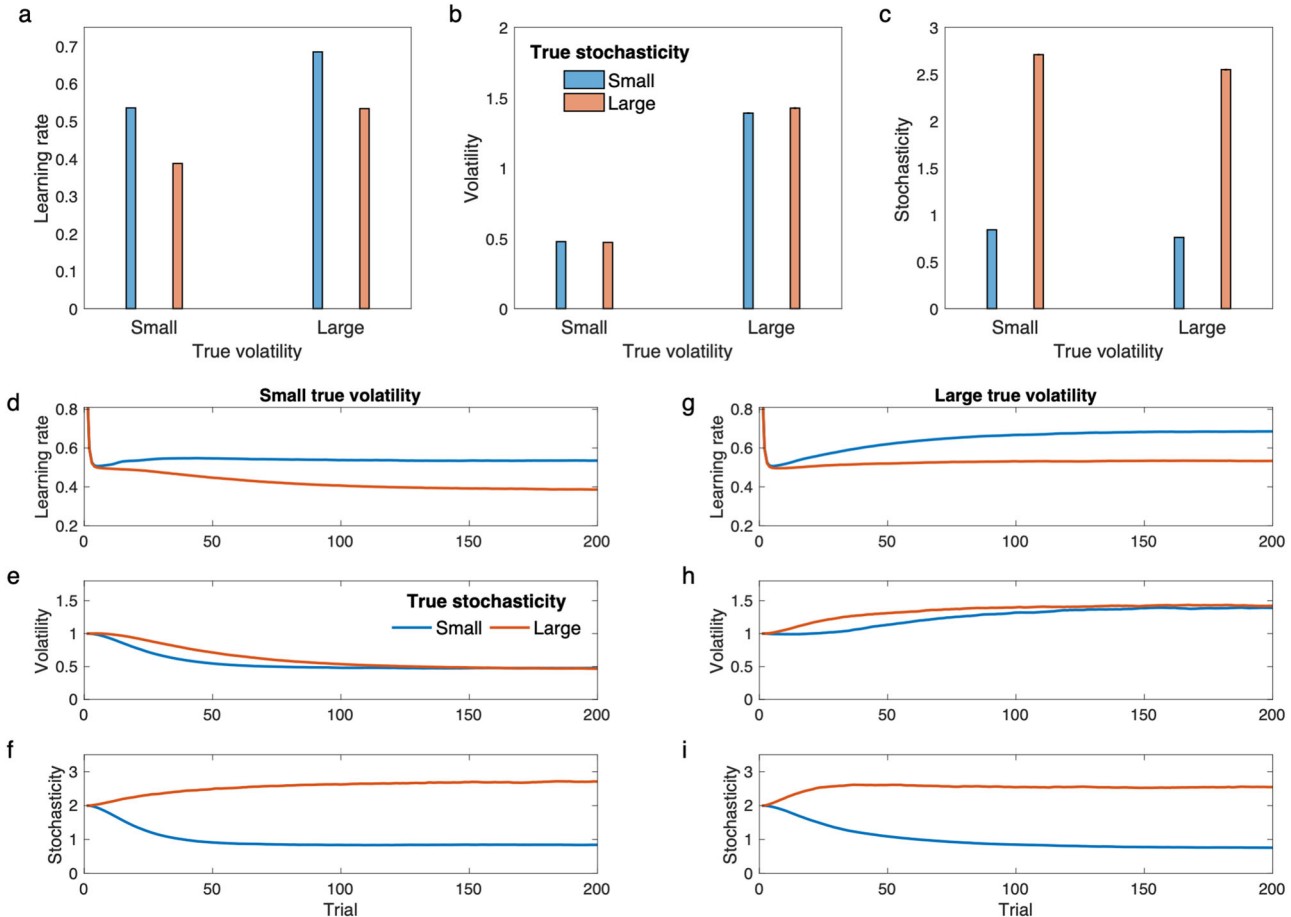

**Fig. 2 Performance of the model in task with constant but unknown volatility and stochasticity parameters.** Outcomes were generated according to the same procedure and parameters as those used in Fig. 1 (see Fig. 1a, b, e.g., outcome time-series seen by the model). **a** Learning rate in the model varies by changes in both the true volatility and stochasticity. Furthermore, these parameters have opposite effects on learning rate. In contrast to volatility, higher stochasticity reduces the learning rate. **b** Estimated volatility captures variations in true volatility. **c** Estimated stochasticity captures variations in the true stochasticity. In **a–c**, average learning rate, estimated volatility and stochasticity in the last 20 trials were plotted over all simulations. **d–f** Learning rate, volatility and stochasticity estimates by the model for small true volatility. **g–i** The three signals are plotted for the larger true volatility. Estimated volatility and stochasticity by the model capture their corresponding true values. Errorbars are standard error of the mean computed over 10,000 simulations and are too small to be visible. See also Supplementary Fig. 3 for further simulation analysis. Source data are provided as Source Data file.

Intriguingly, this situation might in principle arise in neurological damage and psychiatric disorders, if they selectively impact inference about volatility or stochasticity. In that case, the model predicts a characteristic pattern of compensation, whereby learning rate modulation is not merely impaired but *reversed*, reflecting the substitution of volatility for stochasticity or vice versa: a failure of explaining away. Fig. 3 shows this phenomenon in the 2 × 2 design of Fig. 2, with two characteristic lesion models. The key point here is that a lesioned model that does not consider one factor (e.g., stochasticity), inevitably makes systematically incorrect inferences about the other factor too. Importantly, previous models that only consider volatility are analogous to the stochasticity lesion model (Fig. 3b) and, therefore, make systematically erroneous inference about volatility (Fig. 3g) and misadjust learning rate if stochasticity is changing (Fig. 3e). This set of lesioned models provide a rich potential framework for understanding pathological learning in psychiatric and neurologic disorders. Later we show that stochasticity lesion and volatility lesion models explain deficits in learning observed in anxiety and following amygdala damage, respectively. But first, we apply the healthy model to reinterpret some long-standing issues about learning rates in animal conditioning.

**Stochasticity vs. volatility in Pavlovian learning**. Learning rates and their dependence upon previous experience have also been extensively studied in Pavlovian conditioning. In this respect, a distinction emerges between two seemingly contradictory lines of theory and experiment, those of Mackintosh[39] vs. Pearce and Hall[40]. Both of these theories concern how the history of experiences with some cue drives animals to devote more or less "attention" to it. Attention is envisioned to affect several phenomena including not only just rates of learning about the cue but also other aspects of their processing, such as competition between the multiple stimuli presented in compound. Here, to most clearly examine the relationship with the research and models discussed above, we focus specifically on learning rates for a single cue.

The two lines of models start with opposing core intuitions. Mackintosh[39] argues that animals should pay more attention to (e.g., learn faster about) cues that have in the past been more reliable predictors of outcomes. Pearce and Hall[40] argue for the opposite: faster learning about cues that have previously been accompanied by surprising outcomes, i.e., those that have been *less* reliably predictive.

Indeed, different experiments—as discussed below—support either view. For our purposes, we can view these experiments as

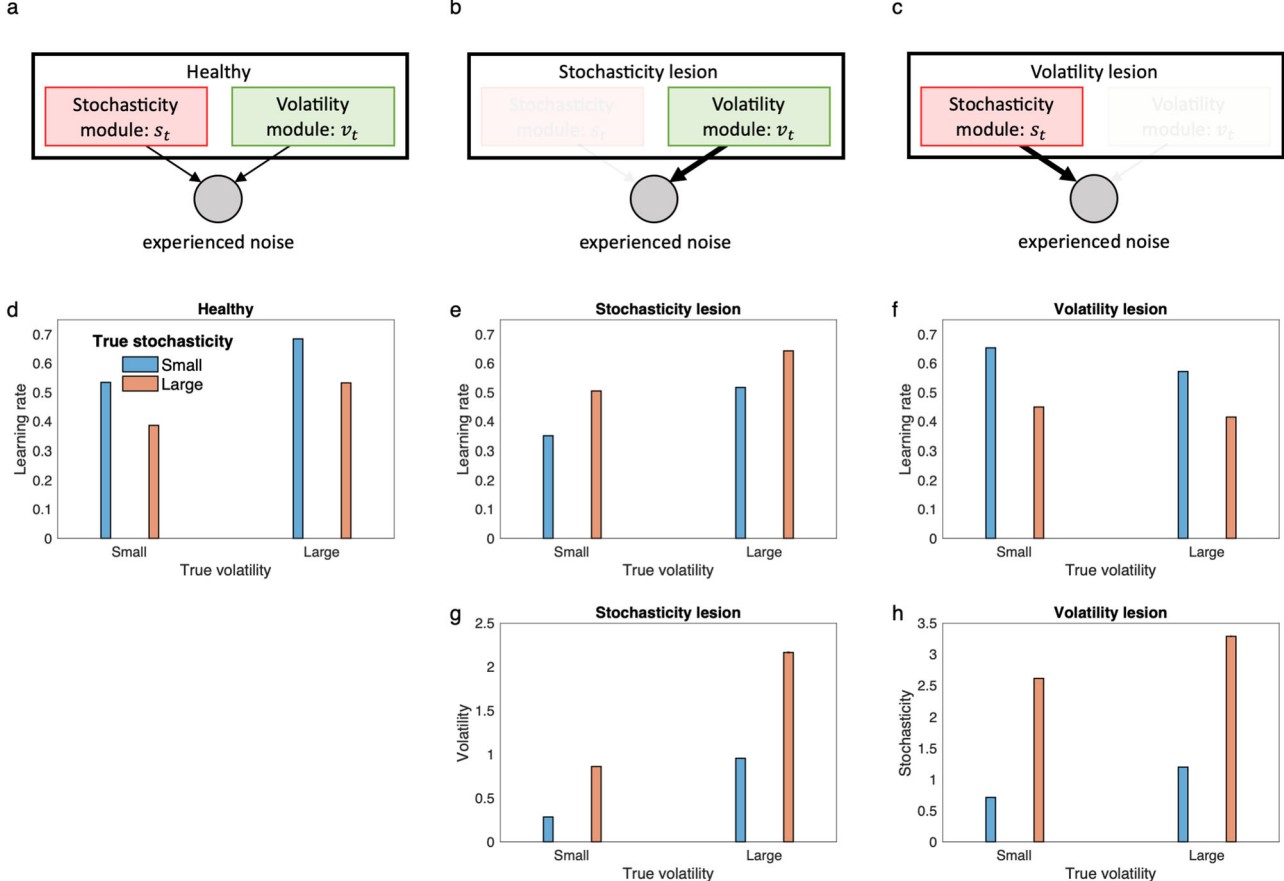

**Fig. 3 Behavior of the lesioned model. a** Stochasticity and volatility module inside the model compete to explain experienced noise. **b–c** Two characteristic lesioned models produce seemingly contradictory behaviors, because if the stochasticity module is lesioned, noise due to stochasticity is misattributed to volatility (**b**), and vice versa (**c**). **d–f** Mean learning rate is plotted for the 2 × 2 design of Fig. 2 for the healthy and lesioned models. For both the lesion models, lesioning does not merely abolish the corresponding effects on learning rate, but reverses them. Thus, the stochasticity lesion model shows elevated learning rate with increases in stochasticity (**e**), and the volatility lesion model shows reduced learning rate with increases in volatility (**f**). This is due to misattribution of the noise due to the lesioned factor to the existing module. **g** The stochasticity lesion model makes erroneous inference about volatility and increases its volatility estimate in higher stochastic environments. **h** The volatility lesion model makes erroneous inference about stochasticity and increases its stochasticity estimate for higher volatile environments. In fact, both the lesion models are not able to distinguish between the volatility and stochasticity and therefore show similar pattern for the remaining module. For the healthy model, volatility and stochasticity estimates are the same as Figs. 2b and 2c, respectively. Simulation and model parameters were the same as those used in Fig. 2. Errorbars reflect standard error of the mean computed over 10,000 simulations and are too small to be visible. Source data are provided as Source Data file.

involving two phases: a pretraining phase that manipulates stochasticity or surprise, followed by a retraining phase to test how this affects the speed of subsequent (re)learning. In terms of our model, we can interpret the pretraining phase as establishing inferences about stochasticity and volatility, which then (depending on their balance) govern learning rate during retraining. On this view, noisier pretraining might, depending on the pattern of noise, lead to either higher volatility and higher learning rates (consistent with Pearce-Hall) or higher stochasticity and lower learning rates (consistent with Mackintosh).

First consider volatility. It has been argued that the Pearce-Hall[40] logic is formalized by volatility-learning models[2,3,12]. In these models, surprising outcomes during pretraining increase inferred volatility and thus speed subsequent relearning. Hall and Pearce[40,53] pretrained rats with a tone stimulus predicting a moderate shock. In the retraining phase, the intensity of the shock was increased. Critically, one group of rats experienced a few surprising "omission" trials at the end of the pretraining phase, in which the tone stimulus was presented with no shock. The speed of learning was substantially increased following the omission trials compared with a control group that experienced

no omission in pretraining. Figure 4a–d shows a simulation of this experiment from the current model, showing that the omission trials lead to increased volatility and faster learning. Note that the history-dependence of learning rates in this type of experiment also rejects simpler models like the Kalman filter, in which volatility (and stochasticity) are taken as fixed; for the Kalman filter, learning rate depends only on the number of pretraining trials but not the particular pattern of observed outcomes. The response probability of the model thus shows the same pattern as response rate for rats (Supplementary Fig. 4).

Next, consider stochasticity. Perhaps the best example of Mackintosh's[39] principle in terms of learning rates for a single cue is the "partial reinforcement extinction effect"[54–56]. Here, for pretraining, a cue is reinforced either on every trial or instead ("partial reinforcement") on only a fraction of trials (Fig. 4e). The number of times that the learner encounters the stimulus is the same for both conditions, but the outcome is noisier for the partially reinforced stimulus. The retraining phase consists of extinction (i.e., fully unreinforced presentations of the cue), which occurs faster for fully reinforced cues even though they had been paired with more reinforcers initially. Our model explains this

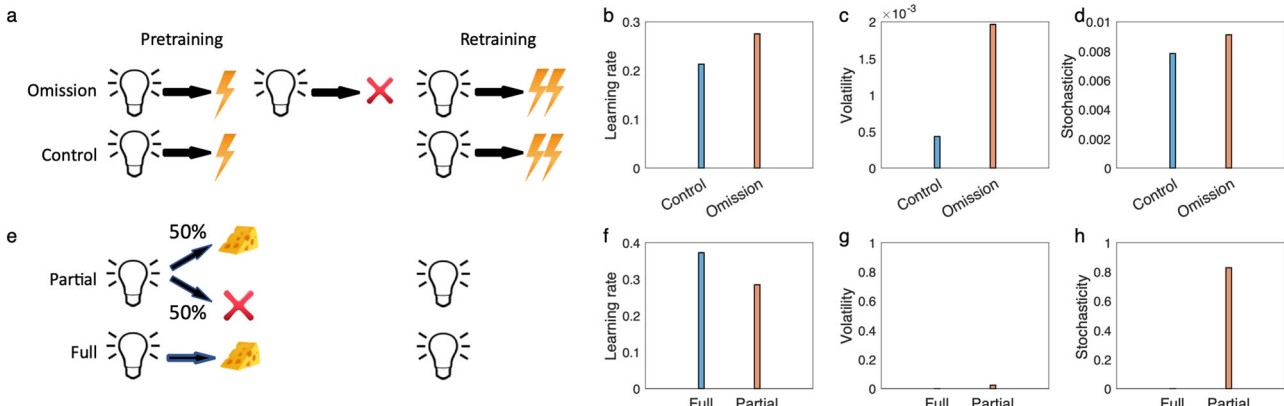

**Fig. 4 The model explains puzzling issues in Pavlovian learning. a–d** Pearce and Hall's conditioned suppression experiment. The design of experiment[51], in which they found that the omission group show higher speed of learning than the control group (**a**) **b** Median learning rate over the first trial of the retraining. The learning rate is larger for the omission group due to increases of volatility (**c**), while stochasticity is similar for both groups (**d**). The model explains partial reinforcement extinction effects (**e–h**). **e** The partial reinforcement experiment consists of a partial condition in which a light cue if followed by reward on 50% of trials and a full condition in which the cue is always followed by the reward. **f** Learning rate over the first trial of retraining has been plotted. Similar to empirical data, the model predicts that the learning rate is larger in the full condition, because partial reinforcements have relatively small effects on volatility (**g**), but it considerably increases stochasticity (**h**). Errorbars reflect standard error of the mean over 40,000 simulations and are too small to be visible. See Supplementary Figs. 4 and 5 for empirical data and corresponding response probability by the model. Source data are provided as Source Data file.

finding (Fig. 4f), because it infers larger stochasticity in the partially reinforced condition, leading to slower learning (Fig. 4g, h). Notably, this type of finding cannot be explained by models, which learn only about volatility[6,11,12]. In general, this class of models mistake partial reinforcement for increased volatility (rather than increased stochasticity), and incorrectly predict faster learning.

Note the subtle difference between the two experiments of Fig. 4. The surprising omission experiment involves stable pretraining prior to omission, then an abrupt shift, whereas pretraining in the partial reinforcement experiment is stochastic, but uniformly so. Accordingly, though both pretraining phases involve increased noise (relative to their controls) the model interprets the pattern of this noise as more likely reflecting either volatility or stochasticity, respectively, with opposite effects on learning rate. Overall, then, these experiments support the current model's suggestion that organisms learn about stochasticity in addition to volatility. Conversely, the models help to clarify and reconcile the seemingly opposing theory and experiments of Mackintosh and Pearce-Hall, at least with respect to learning rates for individual cues. Indeed, although previous work has noted the relationship between Pearce-Hall surprise, uncertainty, and learning rates[2,3,6,12,20,57], the current modeling significantly clarifies this mapping by identifying it more specifically with volatility, as contrasted against simultaneous inference about stochasticity. Meanwhile, while our basic statistical interpretation of the partial reversal extinction effect has been noted before (e.g., by Gallistel and Gibbon[58]), to our knowledge these previous explanations have not reconciled it with the volatility/Pearce-Hall phenomena. Instead, previous work attempting to map Mackintosh's[39] principle onto statistical models (and distinguish it from Pearce-Hall-like effects) has focused on attention and uncertainty affecting cue combination rather than learning rates[2], which is a complementary but separate idea.

**Anxiety and inference about stochasticity vs. volatility.** Lesioned models like the ones in Fig. 3 are potentially useful for understanding learning deficits in psychiatric disorders, for example anxiety disorders, which have recently been studied in the context of volatility and its effects on learning rate[13,20]. These studies have shown that people with anxiety are less sensitive to volatility manipulations in probabilistic learning tasks similar to Fig. 2. Besides learning rates, an analogous insensitivity to volatility has been observed in pupil dilation[13] and in neural activity in the dorsal anterior cingulate cortex, a region that covaried with learning rate in controls[20].

These results have been interpreted in relation to the more general idea that intolerance of uncertainty is a key foundation of anxiety; accordingly, fully understanding them requires taking account of multiple sources of uncertainty[26], including both the volatility and stochasticity. Nevertheless, the primary interpretation of these types of results has been that observed abnormalities are rooted in volatility estimation per se[13,20,26]. Our current model suggests an alternative explanation: that the core underlying deficit is actually with stochasticity, and apparent disturbances in volatility processing are secondary to this, due to their interrelationship.

In particular, these effects and a number of others are well explained by the stochasticity lesion model of Fig. 3b, i.e., by assuming that people with anxiety have a core deficit in estimating stochasticity, and instead treat it as small and constant. As shown in Fig. 5b, this model shows insensitivity to volatility manipulation, but in the model that is actually because volatility is misestimated nearer ceiling due to underestimation of stochasticity. This, in turn, substantially dampens further adaptation of learning rate in blocks when volatility actually increases. The elevated learning rate across all blocks leads to hypersensitivity to noise, which prevents individuals with anxiety from benefitting from stability, as has been observed empirically[20]. In particular, Piray et al.[20] have studied learning in individuals with low- or high- in trait social anxiety using a switching probabilistic task (Supplementary Fig. 6) in which each trial started with a social threatening cue (angry face image). It was found that individuals with high trait anxiety perform particularly worse than controls in stable trials, whereas their performance is generally matched with controls in volatile trials[20] (Fig. 5e). The model shows similar behavior (Fig. 5f).

One key prediction of our model, which differs from a volatility-specific account, is that the learning rate is generally

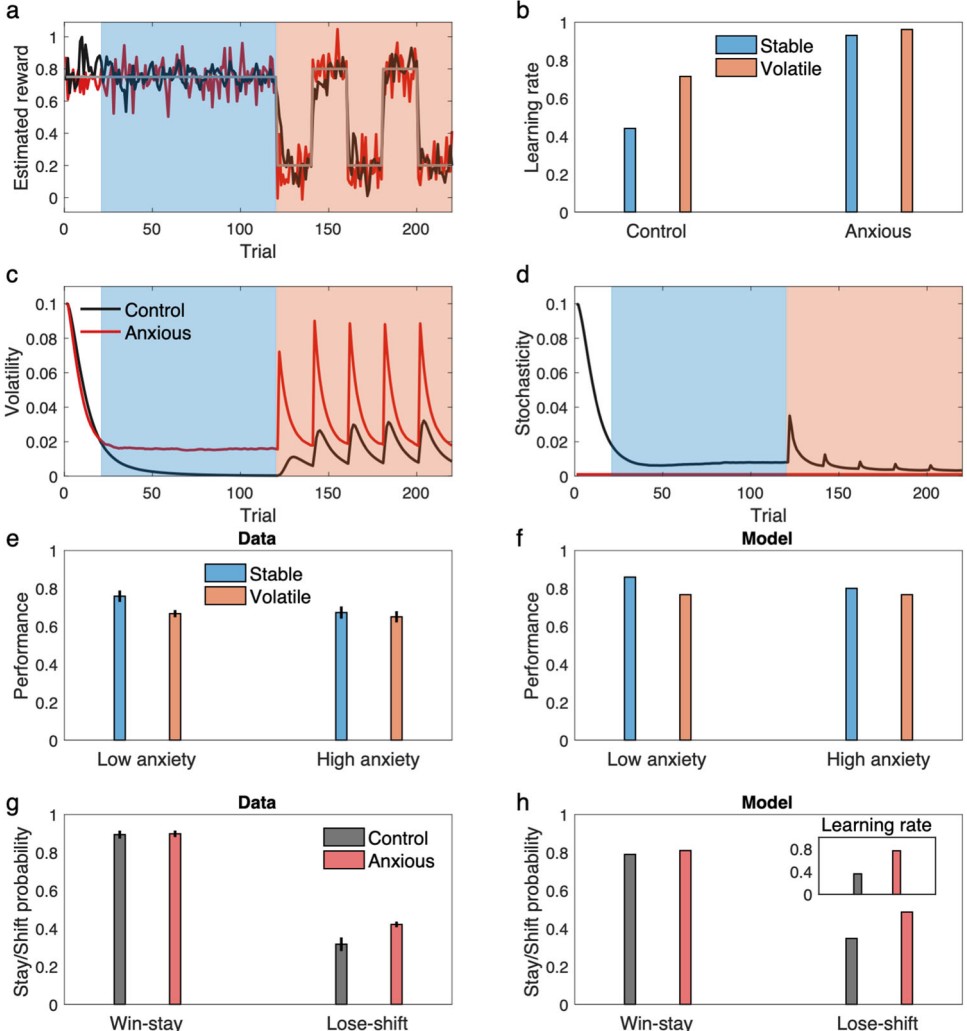

**Fig. 5 The stochasticity lesion model shows a pattern of learning deficits associated with anxiety.** Behavior of the lesioned model as the model of anxiety, in which stochasticity is assumed to be small and constant, is shown along the control model. **a–d** Behavior of the models in the switching task of Fig. 2 is shown. An example of estimated reward by the models shows that the model with anxiety (i.e., the stochasticity lesion model) is more sensitive to noisy outcomes (**a**), which dramatically reduces sensitivity of the learning rate to volatility manipulation in this task (**b**). This, however, is primarily related to inability to make inference about stochasticity, which leads to misestimation of volatility (**c–d**). **e–f** The model explains the data reported by Piray et al.[20], in which the high (social) anxiety group did not benefit from stability as much as the low anxiety group (**e**). The model shows the same behavior (**f**). **g–h** The model explains the data by Huang et al.[32], in which the anxious group showed higher lose-shift behavior compared to the control group (**g**). The model shows the same behavior (**g**), which is due to higher learning rate in the anxious group (inset). Errorbars in (**b**), (**f**), and (**h**) reflect standard error of the mean over 1000 simulations and are too small to be visible. Data in (**e**) are adapted from Piray et al. [20] in which median and standard error of the median are plotted (obtained over $n = 44$ samples). Data in (**g**) are adapted from Huang et al.[32], in which mean and standard error of the mean are plotted (obtained over $n = 122$ independent samples.) Source data are provided as Source Data file.

higher in people with anxiety regardless of volatility manipulation or even in tasks that do not manipulate volatility. In fact, Browning et al.[13] do not find evidence to support this prediction, they do not find a significant overall effect of anxiety on learning rate. Of course, it is important not to interpret null results as evidence in favor of the null hypothesis, since a failure to reject the null hypothesis may reflect insufficient power to detect a true effect. Indeed, in Browning's[13] data, while the effect of anxiety on learning rate was not significant overall or in either condition, the point estimate was largest ($r(28) = 0.26$, $p = 0.16$) in the stable condition, which is also the block that the model predicts the effect should be statistically strongest (because baseline learning rates, absent any effect of anxiety, are lower).

Importantly, other, larger studies provide positive statistical support for the prediction of elevated learning rate with anxiety[31,33,34]. Note that in delta-rule models, behavior under

higher learning rates is closer to win-stay/lose-shift (since higher learning rates weight the most recent outcome more heavily, with full win-stay/lose-shift—dependence only on the most recent outcome—equivalent to a learning rate of 1). Such a strategy has itself been linked to anxiety[33,34]. A notable observation was made in a large ($n = 122$) study by Huang et al.[34], who found people with anxiety show higher win-stay/lose-shift and this effect is driven by higher lose-shift. Figure 5g, h shows results of simulating the proposed model in a task similar to Huang et al.[34] (Supplementary Fig. 6). The model shows the same pattern of behavior, with the additional modulation by win vs. loss captured because any loss is seen as an evidence for volatility and that results in higher learning rate and a contingency switch. The effect is much less salient for win trials because prediction errors are relatively small in those trials, which substantially dampen any effect of learning rate. Across all trials, the

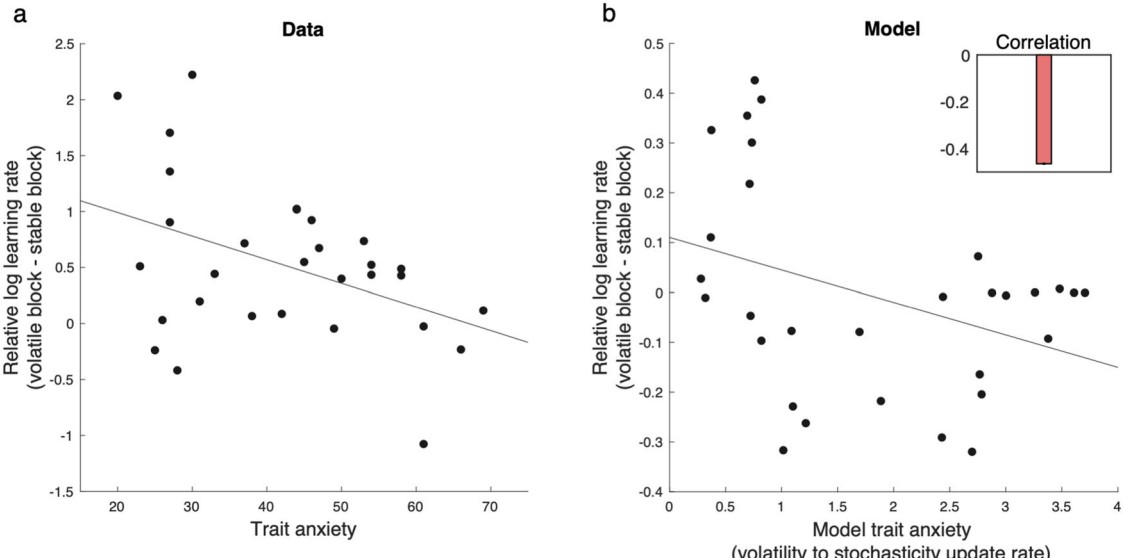

**Fig. 6 The model explains effects of trait anxiety as a continuous index on learning. a** Data by Browning et al.[13] show a significant negative correlation between relative log learning rate and trait anxiety in a probabilistic switching task with stable and volatile blocks. **b** The model shows a similar pattern. The inset shows the median rank correlation between the trait anxiety and the relative learning rate. Model trait anxiety is defined as the ratio of volatility to stochasticity update rates (thus higher if the stochasticity update rate is small). The lesion model of anxiety (Fig. 5) is a special case in which the stochasticity update rate is zero. Errorbars reflect standard error of the median over 1000 simulations and are too small to be visible. Source data are provided as Source Data file.

stochasticity lesion model shows higher learning rate, similar to what Huang et al.[34] found by fitting reinforcement learning models to choice data.

Finally, the lesion model is an extreme case in which a hypothetical stochasticity module is completely eliminated. But this general approach can be extended to less extreme cases in which one module of the model (e.g., stochasticity) has some relative disadvantage in explaining noise. In terms of our model, this can be achieved by having higher update rate parameters for volatility relative to that of stochasticity. These are two main parameters of the model that one can use to explain individual differences across people. For example, the ratio of volatility to stochasticity update rate can be used to capture continuous individual variation in trait anxiety. In this case, the stochasticity lesion model of Fig. 3b is an extreme case of this approach in which the stochasticity update rate is zero (thus the ratio of volatility to stochasticity is infinitely large). We have exploited this approach to simulate a result from Browning et al.[13] concerning graded individual differences in anxiety's effect on learning rate adjustment. In particular, they report (and the model captures; Fig. 6) negative correlation between relative learning rate (volatile minus stable) and trait anxiety in the probabilistic switching task with stable and volatile blocks.

**Amygdala damage and inference about volatility vs. stochasticity.** The opposite pattern of compensatory effects on inference is evidently visible in the effects of amygdala damage on learning. The amygdala plays an important role in associative learning[59,60]. Although some researchers have emphasized a role of the amygdala as a site of association between conditioned- and unconditioned-stimulus in conditioning per se, other authors (drawing on evidence from human neuroimaging work, single-cell recordings, and lesion studies) have proposed that the amygdala is involved in a circuit for controlling or adjusting learning rates[57,60–64]. Most informative from the perspective of our model are lesion studies in rats[61,65,66], which we interpret as supportive an involvement specifically in processing of volatility,

rather than learning rates or uncertainty more generally. These experiments examine a surprise-induced upshift in learning rate similar to the Pearce-Hall experiment from Fig. 4. Lesions to the central nucleus of the amygdala attenuate this effect, suggesting a role in volatility processing. But an important detail of these results with respect to our model's predictions is that the effect is not merely attenuated but reversed. This reciprocal effect supports perhaps the most central feature and prediction of our model that volatility trades off against a (presumably anatomically separate) system for stochasticity estimation.

Figure 7 shows their serial prediction task and results in more detail. Rats performed a prediction task in two phases. A group of rats in the "consistent" condition performed the same prediction task in both phases. The "shift" group, in contrast, experienced a sudden change in the contingency in the second phase. Whereas the control rats showed elevation of learning rate in the shift condition manifested by elevation of food seeking behavior in the very first trial of the test, the amygdala lesioned rats showed the opposite pattern. Lesioned rats showed significantly smaller learning rate in the shift condition compared with the consistent one, a reversal of the surprise-induced upshift.

We simulated the model in this experiment. To model a hypothetical effect of amygdala lesion on volatility inference, we assumed that lesioned rats treat volatility as small and constant. As shown in Fig. 7, the model shows an elevated learning rate in the shift condition for the control rats, which is again due to increases in inferred volatility after the contingency shift. For the lesioned model, however, surprise is misattributed to the stochasticity term as an increase in inferred volatility cannot explain away surprising observations (because it was held fixed). Therefore, the contingency shift inevitably increases stochasticity and thereby decreases the learning rate. Notably, the compensatory reversal in this experiment cannot be explained using models that do not consider both the volatility and stochasticity terms.

A similar pattern of effects of amygdala lesions, consistent with our theory, is seen in an experiment on nonhuman primates. In a recent report by Costa et al.[63], it has been found that amygdala lesions in monkeys disrupt reversal learning with deterministic

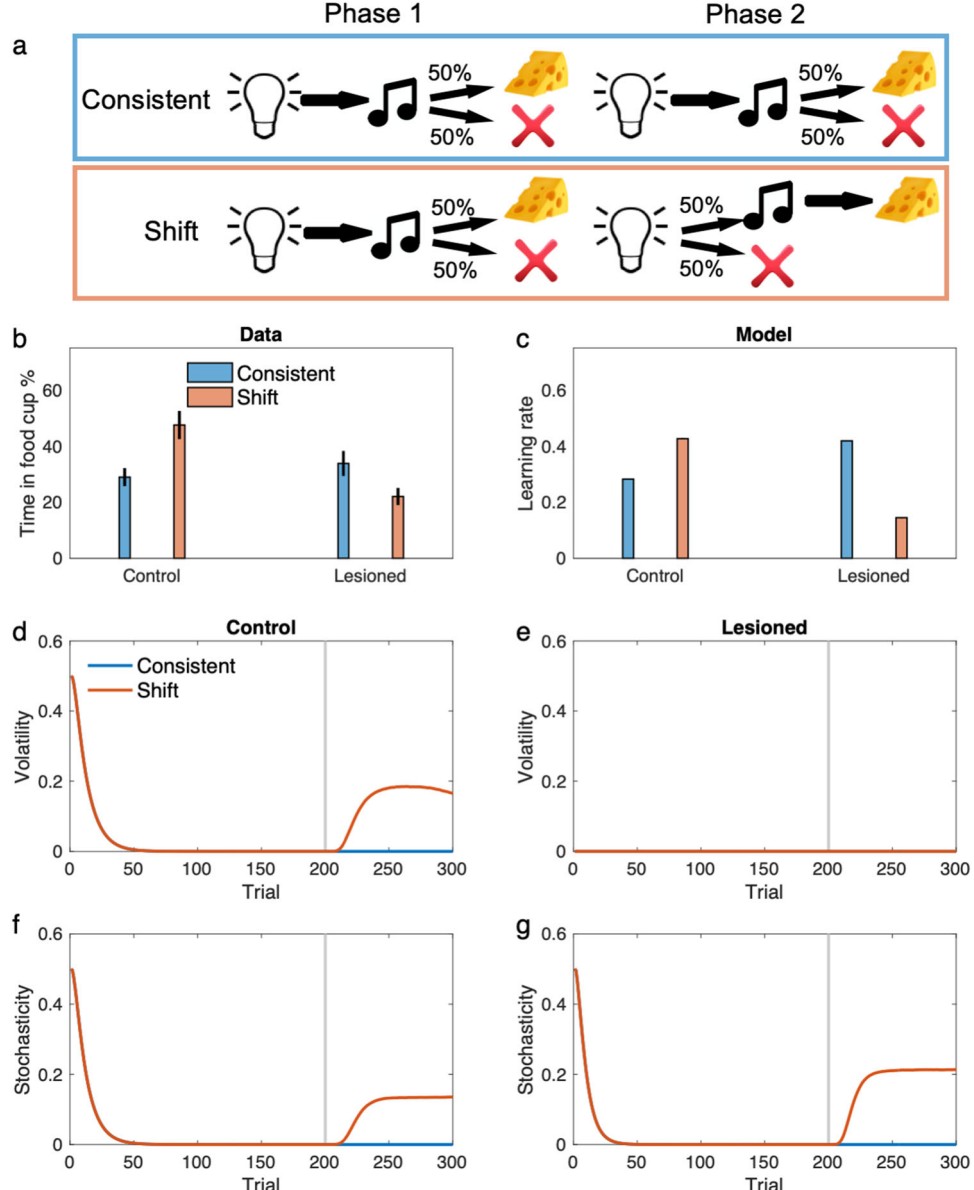

**Fig. 7 The model displays the behavior of amygdala lesioned rats in associative learning. a** The task used for studying the role of amygdala in learning by Holland and Gallagher[59,63,64]. Rats in the "consistent" condition received extensive exposure to a consistent light-tone in a partial reinforcement schedule (i.e., only half of trials led to reward). In the "shift" condition, however, rats were trained on the same light-tone partial reinforcement schedule in the first phase, but the schedule shifted to a different one in the shorter second phase, in which rats received light-tone-reward on half of trials and light-nothing on the other half. **b** Empirical data showed that while the contingency shift facilitates learning in the control rats, it disrupts performance in lesioned rats. **c** learning rate in the last trial of second phase shows the same pattern. This is because the shift increases volatility for the control rats (**d**) but not for the lesioned rats (**e**). In contrast, the contingency shift increases the stochasticity for the lesioned rats substantially more than that for the control rats, which results in reduced learning rate for the lesioned animals (**f-g**). The gray line shows the starting trial of the second phase. Data in (**b**) was originally reported in[63] and reproduced here from[64]. Errorbars in other (**c-g**) reflect standard error of the mean over 40,000 simulations and are too small to be visible. See also Supplementary Table 1. Source data are provided as Source Data file.

contingencies, moreso than a reversal task with stochastic contingencies. This is striking since deterministic reversal learning tasks are much easier. Similar to the previous experiment, our model explains this finding because large surprises caused by the contingency reversal are misattributed to the stochasticity in lesioned animals (because volatility was held fixed), while control animals correctly attribute them to the volatility term (Fig. 8; see Supplementary Fig. 7 for performance of the model and Supplementary Fig. 8 for simulation of the model in all probabilistic schedules tested by Costa et al.[63]). This effect is particularly large in the deterministic case because

the environment is very predictable before the reversal and therefore the reversal causes larger surprises than those for the stochastic one. Similar findings have been found in a study in human subjects with focal bilateral amygdala lesions[67], in which patients tend to show more deficits in deterministic reversal learning than stochastic one. Again, these experimental findings are not explained by a Kalman filter or models that only consider the volatility term.

Overall, then, these experiments support the current model's picture of dueling influences of stochasticity and volatility. Furthermore, the current model helps to clarify the precise role

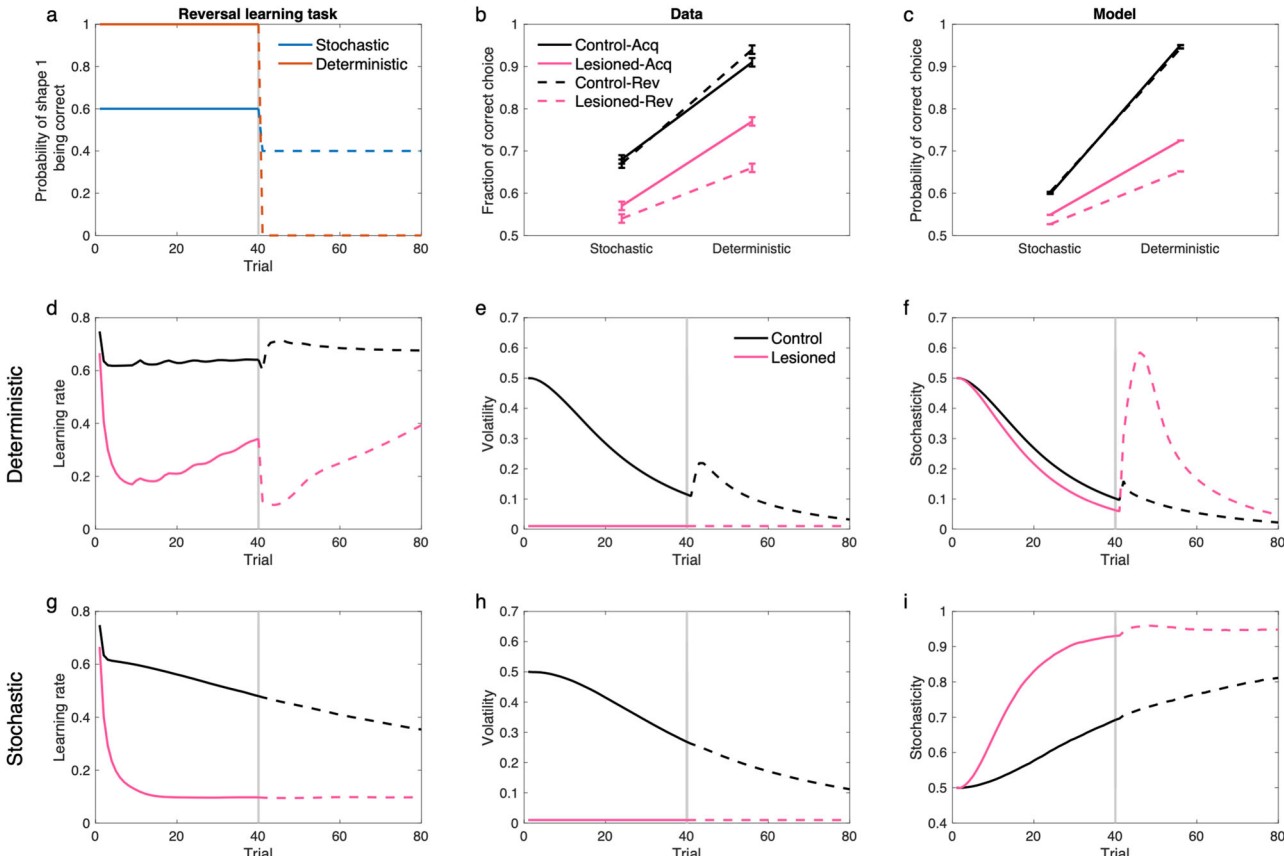

**Fig. 8 The model displays the behavior of amygdala lesioned monkeys in probabilistic reversal learning. a** The probabilistic reversal learning task by Costa et al.[61]. The task consists of 80 trials, in which animals chose one of the two presented shape cues by making a saccade to it and fixating on the chosen cue. A probabilistic reward was given following a correct choice. The stimulus-reward contingency was reversed in the middle of the task (on a random trial between trials 30-50). The task consists of different schedules, but we focus here on 60%/40% (stochastic) and 100%/0% (deterministic), which show the clearest difference in empirical data. **b** Performance of animals in this task. In addition to the general reduced performance by the lesioned animals, their performance was substantially more disrupted in the deterministic- than stochastic-reversal. **c** Performance of the model in this task shows the same pattern. **d–i** Learning rate, volatility and stochasticity signals for the deterministic (**d–f**), and stochastic task (**g–i**). Solid and dashed line are related to acquisition and reversal phase, respectively. Deterministic reversal increases the learning rate in the control animals due to increases in volatility, but not in the lesioned monkeys, in which it reduces the learning rate due to the increase of the stochasticity. The reversal in the stochastic task has very small effects on these signals, because stochasticity is relatively large during both acquisition and reversal. Data in (**b**) are adapted from Costa et al.[61], in which mean and standard error of the mean are plotted. Errorbars in other panels reflect standard error of the mean over 1000 simulations and are too small to be visible. See also Supplementary Fig. 7 for choice time-series and Supplementary Fig. 8 for simulation of the model in all four probabilistic schedules tested by Costa et al.[61] and corresponding empirical data. Source data are provided as Source Data file.

of amygdala in this type of learning, relating it specifically to volatility-mediated adjustments.

## Discussion

A central question in decision neuroscience is how the brain learns from the consequences of choices given that these can be highly noisy. To do so effectively requires simultaneously learning about the characteristics of the noise, as has been emphasized most strongly in a prominent line of work on how the brain tracks the volatility of the environment. Here we revisit this problem for the more realistic case when both volatility and a second noise parameter, stochasticity, must be simultaneously estimated.

While various experiments have, mostly separately, shown that humans can adjust learning rates in response to manipulations of either type of noise, models of how they do so have focused primarily on estimating either parameter while taking the other as known. This skirts the more difficult problem of distinguishing types of noise. To solve this problem and investigating its

consequences for learning, we built a probabilistic model for learning in uncertain environments that tracks volatility and stochasticity simultaneously. Using this model to simulate a number of experiments across conditioning, psychiatry and lesion studies, we show a consistent theme whereby the interdependence of inference about these two noise parameters gives rise to patterns of effects that could not be appreciated in previous models that considered estimating either type of noise separately.

The importance of dissociating these forms of noise, and some aspects of their interaction, have been noted previously. For instance, Pulcu and Browning[26] emphasize the inadequacy of existing experiments for dissociating volatility vs. stochasticity learning, and raise the possibility that in principle, people might confuse them. In Nassar et al.'s study[25], the volatility-like hazard rate parameter (though viewed from the model's perspective as fixed and known) is fit as a per-subject free parameter construed as an individual difference. The empirical and model-fitting results showcase a dependence of the (inferred) stochasticity parameter upon the (fit/known) hazard rate, consistent with the bidirectional pattern of interdependence we posit. Building on all

these ideas, we build and simulate a model to showcase the potential interdependence between these two types of inference across a range of situations. One important caveat, given the range of applications we consider, is that we abstract away details of the many individual studies to emphasize their parallelism with respect to our main point of interest. Thus, for instance, we neglect valence-dependent modulation of learning which is likely an additional dimension important both in anxiety[20,26,31,38] and in studies of amygdala[63]. Relatedly, as our goal is to showcase the range of situations in which parallel issues may arise, we acknowledge that different explanations may exist for many individual results.

Our work builds most directly on a rich line of theoretical and experimental work on the relationship between the volatility and learning rates[6,8,13,15,27,68,69]. There have been numerous reports of volatility effects on healthy and disordered behavioral and neural responses, often using a two-level manipulation of volatility like that from Fig. 5a[6,8,10,13–23,38]. Our modeling suggests that it will be informative to drill deeper into these effects by augmenting this task to cross this manipulation with stochasticity so as more clearly to differentiate these two potential contributors[24]. For example, in tasks that manipulate (and models that consider) only volatility, it can be seen from Eqs. (1–4) that the timeseries of several quantities all covary together, including the estimated volatility $v_t$, the posterior uncertainty $w_t$, and the learning rate $\alpha_t$. It can therefore be difficult in general to distinguish which of these variables is really driving tantalizing neural correlates related to these processes, for instance in amygdala and dorsal anterior cingulate cortex[6,57]. The inclusion of stochasticity (which increases uncertainty but decreases learning rate) would help to drive these apart.

Indeed, another related set of learning tasks considered prediction of continuous outcomes corrupted by stochasticity, i.e., additive Gaussian noise[7,9,25,45,52], which could provide another foundation for factorial manipulations of the sort we propose. Indeed, a number of these studies (complementary to the volatility studies) included multiple levels of stochasticity and showed learning rate effects[7,9,25,52,70]. The models used in these studies have largely used a complementary simplification to the volatility one: they estimate stochasticity, but conditional on a known value for the hazard rate (equivalent to volatility). Interestingly, rather than overall adjustment to noise statistics, these studies more explicitly emphasized the detection of *discrete* changes in the environment and the resulting local adjustments of the learning rate. From a modeling perspective, inference under change at discrete changepoints (occurring at some hazard rate) raises issues quite analogous to change due to more gradual diffusion (with some volatility). Thus, in practice it has been common and effective to apply models for one sort of change to tasks actually involving the other[6,11,12], a substitution also in part licensed by approximate models of changepoint detection that (as with the volatility models and the Kalman filter for continuous change) also reduce learning to error-driven updates with a time-varying learning rate[25,71]. Thus, although we build the current work on a generative model with continuous rather than abrupt changepoints, we do not mean this as a substantive claim, as we expect our main substantive points (concerning the inference about noise vs. change hyperparameters) would play out analogously in other variants. In any case, research into the neural substrates of changepoint detection is highly relevant to the change problem conceived in terms of volatility as well (see[10] for a recent review).

The current framework's tendency to elide the distinction between discrete and continuous change (but distinguish both from stochasticity) is also the basis of an important, but subtle, distinction from another prominent dichotomy previously proposed that between "expected" and "unexpected" types of

uncertainty[72,73]. While it might appear that these categories correspond, respectively, to stochasticity and volatility as we define them, that is not actually the case. Formally, this is because the Dayan and Yu model (in its most detailed form, Dayan and Yu[72]) arises from a Kalman filter augmented with additional discrete changepoints: i.e., both the diffusion and jumps. The focus of that work was distinguishing the special effects of surprising jumps ("unexpected uncertainty"), which were hypothesized to recruit a specialized neural interrupt system. Meanwhile, all other uncertainty arising in the baseline Kalman filter (i.e., that from both stochasticity and volatility; the posterior variance $w_t$ in Eq. 4) is lumped together under "expected uncertainty." That said (although we see this as a misreading of the earlier work) our impression is that later authors' use of these terms actually tends to comport more with our distinction than the original definition[26], i.e., to take unexpected and expected uncertainty as synonymous with volatility and stochasticity as we define them. In any case, Yu and Dayan did not consider the problem considered here, of estimating the noise hyperparameters for learning under uncertainty.

The most important feature of our model is the competition it induces between the volatility and stochasticity to "explain away" surprising observations. This leads to a predicted signature of the model in cases of lesion or damage affecting inference about either type of noise: disruption causing neglect of one of the terms leads to overestimation of the other term. For example, if a module responsible for volatility learning were disrupted, the model would overestimate stochasticity, because surprising observations that are due to volatility would be misattributed to the stochasticity. This allowed us to revisit the role of amygdala in associative learning and explain some puzzling findings about its contributions to reversal learning.

Similar explanations grounded in the current model may also be relevant to a number of psychiatric disorders. Abnormalities in uncertainty and inference, broadly, have been hypothesized to play a role in numerous disorders, including especially anxiety and schizophrenia. More specifically, abnormalities in volatility-related learning adjustments have been reported in patients or people reporting symptoms of several mental illnesses[13,14,16–23,38]. The current model provides a more detailed potential framework for better dissecting these effects, though this will ideally require a new generation of experiments manipulating both factors.

In the present work, we have developed these ideas mostly in terms of pathological decision making in anxiety, which is one of the areas where earlier work on volatility estimation has been strongest and where further refinement using our theory seems most promising[32,35,37,74]. We considered an account by which individuals with anxiety systematically misidentify outcomes occurring due to chance (stochasticity) as instead a signal of change (volatility)[34]. This account offers a contrary interpretation of a pattern of effects that had been taken to indicate that volatility sensitivity is instead deficient in anxiety[26]. Although some null effects in the study of Browning et al.[13] do not support this account, we view it as an overall better account of the pattern of data across several studies[20,31,33,34]. Our account is also broadly consistent with studies suggesting that individuals with anxiety might feel overwhelmed when faced with uncertainty[36] and fail to make use of long-term statistical regularities[34]. These are also hypothesized to be related to symptoms of excessive worry[29,30]. Misestimating stochasticity—moreso than volatility—also seems consonant with the idea that individuals with anxiety tend to fail to discount negative outcomes occurring by chance (i.e., stochasticity) and instead favor alternative explanations like self-blame[75]. This hypothesis is also consistent with the observation that acquisition of fear conditioning tends to be enhanced in

individuals with anxiety[76,77]. Finally, although a simple increase in learning rate seems harder to reconcile with generally slower extinction of Pavlovian fear learning in anxiety[76], this probably reflects the well-known fact that extinction is not simply unlearning of the original associations, but instead is dominated by additional processes[78,79]. This includes in particular statistical inference about latent contexts[5], which is likely to be affected by both stochasticity and volatility in ways that should be explored in future work.

More generally, this modeling approach, which quantifies misattribution of stochasticity to volatility and vice versa, might be useful for understanding various other brain disorders that are thought to influence processing of uncertainty and have largely been studied in the context of volatility in the past decade[14,16,17,19,21,22,27,28]. As another example, positive symptoms in schizophrenia have been argued to result from some alterations in prior vs likelihood processing, perhaps driven by abnormal attribution of uncertainty (or precision) to top-down expectations[80]. But different such symptoms (e.g., hallucinations vs. delusions) manifest in different patients. One reason may be that these relate to disruption at different levels of a perceptual-inferential hierarchy, i.e., with hallucination vs. delusion reflecting involvement of more or less abstract inferential levels, respectively[81–83]. In this respect, the current model may provide a simple and direct comparative test, since stochasticity enters at the perceptual, or outcome, level (potentially associated with hallucination) but volatility acts at the more abstract level of the latent reward (and may be associated with delusion; see Fig. 1).

Our work also touches upon a historical debate in the associative learning literature about the role of outcome stochasticity (i.e., in our terms, noise) in learning. One class of theories, most prominently represented by Mackintosh[39], proposes that attention is preferentially allocated to cues that are most reliably predictive of outcomes, whereas Pearce and Hall[62] suggest the opposite that attention is attracted to surprising misprediction. We address only a subset of the experimental phenomena involved in this debate (those involving learning rates for cues presented alone), but for this subset we offer a very clear resolution of the apparent conflict. Our approach and goals also differ from classic work in this area. A number of important models of attention in psychology also attempt to reconcile these theories by providing more phenomenological models that hybridize the two theories to account for various and often paradoxical experimental work[84–87]. Our goal is different and is more descended from a tradition of normative theories that provide a computational understanding of psychological phenomena from first principles by first addressing what is the computational problem that the corresponding neural system is evolved to solve[2,88].

Any probabilistic model relies on a set of explicit assumptions about how observations have been generated, i.e., a generative model, and also an inference procedure to estimate the hidden parameters that are not directly observable. Such inference algorithms typically reflect some approximation strategy because exact inference is not possible for most important problems, including our generative model (Fig. 1). In previous work in this area, we and others have relied on variational approaches to approximate inference, which factors difficult inference problems into smaller tractable ones, and approximates the answer as though they were independent[11,12]. Interestingly, although one of the most promising successes of this approach in neuroscience has been in hierarchical Kalman filters with volatility inference, we found it difficult to develop an effective variational filter for the current problem, when stochasticity is unknown. The core problem, in our hands, was that effective explaining away between the two noise types was difficult to achieve using simplified variational posteriors that omitted aspects of their mutual dependency.

Interestingly, there are other algorithms that, in principle, address similar learning problems. These include using an explicitly variational approach extending the HGF (code is publicly available as hgf_jget in the TAPAS toolbox[89], but has not been documented or tested in published articles), augmenting the variational HGF with mixture models[43], an analogous simplified learning rule based more on neural considerations[44], and an exact model for tracking hazard rates under a particular case of changepoint detection[45]. While these have not yet been applied to the full range of problems we investigate here, we suspect that future work investigating the approximate approaches will find challenges in explaining away. In any case, in the current modeling, we have adopted a different estimation method based on Monte Carlo sampling, in particular a variant of particle filtering that preserves many of the advantages of variational methods by incorporating exact conditional inference for a subset of variables[49]. The inference model employed here combines Kalman filtering for estimation of reward rate[41] conditional on the volatility and stochasticity, with particle filtering for inference about these[50]. One drawback of the particle filter, however, is that it requires tracking a number of samples on every trial. In practice, we found that a handful (e.g., 100) of particles results in a sufficiently good approximation.

Finally, in this study, we only modeled the effects of volatility and stochasticity on learning rate. However, uncertainty affects many different problems beyond learning rate, and a full account of how subjects infer volatility and stochasticity (and how these, in turn, affect uncertainty) may have ramifications for many other behaviors. Thus, there have been important statistical accounts of a number of such problems, but most of them have neglected either stochasticity or volatility, and none of them have explicitly considered the effects of *learning* the levels of these types of noise. These problems include cue- or feature-selective attention[2]; the explore-exploit dilemma[90,91]; and the partition of experience into latent states, causes or contexts[5,79,92]. The current model, or variants of it, is more or less directly applicable to all these problems and should imply predictions about the effects of manipulating either type of noise across many different behaviors.

## Methods

**Description of the model**. Recall that outcome on trial $t$, $o_t$, in our model depends on three latent variables, the reward rate, stochasticity and volatility. The reward rate on trial $t$, $x_t$, has Markov-structure dynamics:

$$x_t = x_{t-1} + e_t, \qquad (5)$$

where $e_t$ is a (zero-mean) Gaussian noise with variance given by volatility. Therefore, we have:

$$p(x_t|x_{t-1}, v_t) = N(x_t|x_{t-1}, v_t), \qquad (6)$$

where $v_t$ is volatility. We define the inverse volatility, $z_t = v_t^{-1}$, which is the preferred formulation here as it has been used in previous studies for its analytical plausibility[12]. Outcomes were generated based on the reward rate and stochasticity according to a Gaussian distribution:

$$p(o_t|x_t, s_t) = N(o_t|x_t, s_t), \qquad (7)$$

where $s_t$ is the stochasticity with $y_t = s_t^{-1}$.

For volatility and stochasticity, we assumed a multiplicative noise on their inverse, which is an approach that has been shown to give rise to analytical inference when considered in isolation (but not here)[93,94]. Specifically, the dynamics over these variables are given by $z_t = \eta_v^{-1} z_{t-1} \epsilon_t$, where $0 < \eta_v < 1$ is a constant and $\epsilon_t$ is a random variable in the unit range with a Beta-distribution $p(\epsilon_t) = B(\epsilon_t, |, 0.5\eta_v(1 - \eta_v)^{-1}, 0.5)$. Note that the conditional expectation of $z_t$ is given by $z_{t-1}$, because $E[\epsilon_t] = \eta_v$. We assume a similar and independent dynamics for $y_t$ parametrized by the constant $\eta_s$: $y_t = \eta_s^{-1} y_{t-1} \varepsilon_t$, in which $\varepsilon_t$ has a similar distribution to $\epsilon_t$ parametrized by $\eta_s$.

In our implementation, we parametrized the model with $\lambda_v = 1 - \eta_v$ and $\lambda_s = 1 - \eta_s$, respectively. This is because these parameters can be interpreted as the

update rate for volatility and stochasticity, respectively. In other words, larger values of $\lambda_v$ and $\lambda_s$ result in faster update of volatility and stochasticity, respectively. Intuitively, this is because a smaller $\lambda_v$ increases the mean of $\epsilon_t$ and results in a larger update of $z_t$. Since volatility is the inverse of $z_t$, therefore, smaller $\lambda_v$ results in slower update of volatility. This has been formally shown in our recent work[12]. In addition to these two parameters, this generative process depends on initial value of volatility and stochasticity, $v_0$ and $s_0$.

For inference, we employed a Rao-Blackwellised Particle Filtering approach[49], in which the inference about $v_t$ and $s_t$ were made by a particle filter[50] and, conditional on these, the inference over $x_t$ was given by the Kalman filter (i.e., Equations (1–4)). The particle filter is a Monte Carlo sequential importance sampling method, which keeps track of a set of particles (i.e., samples). The algorithm performs three steps on each trial. First, in a prediction step, each particle is transitioned to the next step based on the generative process. Second, weights of each particle are updated based on the probability of observed outcome:

$$b_t^l \propto N(o_t | m_{t-1}^l, w_{t-1}^l + v_t^l + s_t^l), \qquad (8)$$

where $b_t^l$ is the weight of particle $l$ on trial $t$, $m_{t-1}^l$ and $w_{t-1}^l$ are estimated mean and variance by the Kalman filter on the previous trial (Eqs. 1–4), and $v_t^l$ and $s_t^l$ are volatility and stochasticity samples (i.e., the inverse of $z_t^l$ and $y_t^l$). In this step, particles were also resampled using the systematic resampling procedure if the ratio of effective to total particles falls below 0.5. In the third step, the Kalman filter was used to update the mean and variance. In particular, for every particle, Eqs. 1–4 were used to define $\alpha_t^l$ and update $m_t^l$ and $w_t^l$ for every particle. Learning rate and estimated reward rate on trial $t$ was then defined as the weighted average of all particles, in which the weights were given by $b_t^l$. We have used particle filter routines implemented in MATLAB.

Finally, we should note that our results are not dependent on the specific generative process that we have assumed here. In particular, it is possible to define a generative process that diffuses according to Gaussian noise. In such a generative model, random variables related to volatility and stochasticity diffuse according to independent Gaussian random walks:

$$p(z_t | z_{t-1}) = N(z_t | z_{t-1}, \sigma_v^2), \qquad (9)$$

$$p(y_t | y_{t-1}) = N(y_t | y_{t-1}, \sigma_s^2), \qquad (10)$$

where volatility and stochasticity are respectively defined as $v_t = \exp(z_t)$ and $s_t = \exp(y_t)$, and $\sigma_v$ and $\sigma_s$ are model parameters that play analogous role as $\lambda_v$ and $\lambda_s$ above, respectively. The reward rate and outcomes are then generated based on the same process as the previous generative model (Eqs. 5–7). Inference about reward rate also remains the same (Eqs. 1–4). Our simulations show that, as long as the particle filter was used for inference about volatility and stochasticity, such a process can successfully recover true unknown volatility and stochasticity (Supplementary Fig. 3).

**Simulation details**. In simulations related to Figs. 1–3, timeseries were generated according to the Markov random walk with constant volatility and stochasticity. For these simulations, we assumed $\lambda_v = 0.1$ and $\lambda_s = 0.1$; $v_0 = 1$ (average over small and large true volatility) and $s_0 = 2$ (average over small and large true stochasticity). For lesioned models in Fig. 3, the corresponding lesion variable was assumed to be fixed at its initial value throughout the task.

For the conditioned suppression experiment presented in Fig. 4, the weak and strong shock was 0.3 and 1, respectively, plus a small noise with variance of $10^{-2}$. The noise for the partial reinforcement experiment was assumed to be $10^{-4}$. 100 trials were used for training. We assumed 5 omission trials for the omission condition of conditioned suppression experiment. Model parameters in Fig. 4 were $\lambda_v = \lambda_s = 0.2$, and $v_0 = s_0 = 0.1$. For corresponding Supplementary Figs. 4–5, the response probability of the model was calculated based on a softmax with a decision noise parameter of 5.

Reward rate in Fig. 5a was 0.8 in the stable block and switching between 0.25 and 0.75 in the volatile block with the outcome variance of 0.01. For simulations presented in Fig. 5a–d, model parameters were similar to those used for simulations in Fig. 4 and the stochasticity for the lesioned models was assumed to be 0.001 (note that outcomes were not binary in this simulation). Volatile condition in Fig. 5e, f was defined as trials with no contingency switch in their preceding 10 trials, similar to Piray et al.[20]. For the simulation presented in Fig. 5h, we followed Huang et al.[34] who fitted a number of reinforcement learning models to choice data, in which they simplified the task to its core features that are directly related to reinforcement learning. Furthermore, we made a further simplification here by considering only two choices. Probability of reward in tasks by Piray et al. and Huang et al. presented in Fig. 5 are plotted in Supplementary Fig. 6. Outcome variance was assumed to be 0.01. Model parameters were similar to those used for simulations in Fig. 4. The stochasticity for the lesioned models in these two simulations were assumed to be 0.05. For simulating choice, we used the softmax with a decision noise of 3.

In Fig. 6, the task was a probabilistic switching task with stable and volatile blocks similar to the task of Browning et al.[13]. Outcome variance was assumed to be 0.01. The median correlation presented in the inset of Fig. 6b is the Spearman rank correlation across 1000 sets of simulations. For each set, 30 artificial subjects

were generated, which only differed in their volatility- and stochasticity-update rate parameters. To have a relatively uniform model trait anxiety (i.e., volatility to stochasticity update rate), every set was further divided to 3 subsets (each containing 10 artificial subjects), in which the mean of model trait anxiety was 0.5, 1, 3, respectively. We further ensured that the model trait anxiety is greater than 0.26 and smaller than 4. These values were chosen to relatively reflect the distribution of trait anxiety in Browning et al.'s[13] data (Fig. 6a). Furthermore, the volatility update rate was drawn randomly between 0 and 0.2 and the stochasticity update rate was calculated according to the model trait anxiety. A fixed and small initial volatility and stochasticity was used for all artificial subjects ($v_0 = s_0 = 0.001$).

For simulating the experiment in Fig. 7, reward timeseries was generated with a very small outcome variance, $10^{-6}$. Here, the model was trained to predict both the tone (given the light) and the reward (given the light) on every trial. Model parameters were $\lambda_v = \lambda_s = 0.2$, and $v_0 = s_0 = 0.5$. Volatility was assumed for the lesioned model to be small and fixed (0.25e-6). Figure 7b shows the average learning rate on the last trial of phase 2 (i.e., the first trial of the test) for the first cue across all simulations. Figure 7c–f shows volatility and stochasticity signals for the first cue. For simulation of the reversal task in Fig. 8, a small outcome variance similar, $10^{-6}$, was used for generating outcomes. Model parameters were the same as those used in Fig. 7. Volatility for the lesioned model was assumed to be small and fixed at 0.01. We have used softmax with a decision noise parameter as the choice model. We assumed that the decision noise parameter is 3 and 1 for the control and lesioned animals, respectively. These parameters were used to reproduce the general reduction of performance in lesioned animals, which is independent of the difference between the deterministic vs stochastic task in the two groups explained by our learning model.

For Supplementary Fig. 2, we followed the design of the prediction task by Nassar et al.[25]. The timeseries were generated according to a hidden reward rate plus a noise term, in which the variance of noise was either one (small) or nine (large). The reward rate was subject to a random jump in the range of 0–10. Change points occurred after at least five trials plus a random draw according to an exponential distribution with a rate of 0.05 (i.e., mean 20). The initial volatility and stochasticity were both assumed to be five (average over small and large true stochasticity). We further assumed that $\lambda_v = 0.4$ and $\lambda_s = 0.2$, which reflects the instructions given to subjects about possible jumps in the underlying reward rate. Simulation and model parameters in Supplementary Figure 1 were the same as those in Fig. 5a. For all simulations, we have assumed initial reward rate prior to be a Gaussian with mean 0 and variance 100 (Figs. 2–3 and Supplementary Figure 3) or 1 (all other simulations). Simulations were repeated sufficiently to have negligible sampling error (i.e. invisible standard error of the mean). Thus, simulations presented in Figs. 1, 2, 3 and Supplementary Fig. 3 were repeated 10,000 times; conditioning simulations presented in Figs. 4 and 7 were repeated 40000 times; and all other simulations were repeated 1000 times. All simulations were conducted with 100 particles.

**Reporting Summary**. Further information on research design is available in the Nature Research Reporting Summary linked to this article.

## Data availability
Simulation data are publicly available at https://doi.org/10.5281/zenodo.5526668[95]. Data by Piray et al.[20] presented in Fig. 5e are publicly available at https://github.com/payampiray/piray_etal_2019_JNeurosci. Data by Browning et al.[13] plotted in Fig. 6a are publicly available as a Source Data file to the corresponding paper. Source data are provided with this paper.

## Code availability
All simulations were conducted using custom code written in MATLAB (2018a). Codes are available at https://doi.org/10.5281/zenodo.5526668[95].

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

## Acknowledgements
We thank Sam Zorowitz, Peter Dayan, Yoel Sanchez Araujo, and Guillermo Horga for helpful discussions. This work was supported by grants IIS-1822571 from the National Science Foundation, part of the CRNCS program, and 61454 from the John Templeton Foundation.

## Author contributions
P.P. and N.D.D. designed the study and wrote the manuscript. P.P. performed analyses.

## Competing interests
The authors declare no competing interests.
