## [Peer Review File · Nature Communications]

A model for learning based on the joint estimation of
stochasticity and volatilityREVIEWER COMMENTS

Reviewer #1 (Remarks to the Author):

Piray and Daw develop a particle filter-based approximate inference model for learning in environments with unknown “volatility” and “unpredictability”. They apply the model to explain seemingly incongruent results from conditioning in rodents, human behavior across multiple tasks, changes in behavior that emerge after amygdala lesions, and individual differences in learning that occur with anxiety.

This paper is impressive in that its scope. It attempts to capture data from three species in a wide range of task designs. In my opinion, a primary advantage of models is that they provide a way to generalize across the task specifics – and I think this paper makes an impressive contribution in that direction. However, I have a large number of concerns about the paper, many of which are probably consequences of the strategy taken by the authors to model many things, but not model anything in detail. My hope is that the authors are able to address these concerns, as I think if adequately revised the paper could provide a very nice contribution to the field.

Major concerns:

The authors motivate the novelty of their work based on a new model where volatility and unpredictability are learned simultaneously from experience. I would argue that they are not the first to do this – for example, Wilson 2010 provides an algorithm for optimal inference in environments with changepoints and unknown variance – that maps qualitatively onto the constructs explored here. Nor are the authors the first to explore the implications for the interaction between volatility and unpredictability estimation either theoretically (Yu & Dayan 2005) or in relation to individual differences in behavior (eg. noise estimation model from Nassar 2010). The implications for conditioning have also been described previously to some extent (Courville & Daw 2006) – and the implications for anxiety have also been discussed (Pulco & Browning 2019). What has never been done previously, that is done in this manuscript, is to systematically apply one model to such a broad array of task findings across experimental designs and species. I think that is really valuable, and in my opinion it is what makes this work appropriate for a broad readership. But I think reframing could be necessary to clarify exactly what this work provides beyond the work listed above.

My second major concern is that while the breadth of modeling applied here is commendable, the depth of explanation for each condition leaves something to be desired. Most of the major concerns below are different specific cases of this issue.

For example, figure 3 models a task with rich behavioral dynamics that have been extremely well characterized... but the only thing that is modeled is the difference in average learning rate under high and low noise... conditions. While this is a feature of human behavior in that task, it is a minor one, with the more impressive features being the scaling of learning rate as a function of error magnitude and noise, and the scaling of learning rate with the number of trials since the last change. If the authors wish to make the claim that the discrete transitions in that task can be well modeled by the drifting process in their model it seems imperative to show that the model captures these core features of behavior.

For all simulations, but particularly the conditioning simulations, I found it very difficult to understand exactly what is being shown to the model (ie. how many trials per phase, how were values assigned to each condition). This makes interpretation of the model “behavior” difficult. But from what I can see, I suspect that this model diverges pretty systematically from the rodent behavior in that it uses learning rates that are extremely high. I know the data from several of the cited conditioning experiments quite well, and none that I know of show anything that looks like one shot learning. Acquisition and extinction curves are slow -- and based on the learning rates presented I would guess that this model vastly underestimates the number of trials that would be necessary to acquire/extinguish conditioning behavior. Claims could be strengthened by showing that the model

predicts some aspects of behavior beyond the binary higher/lower comparison – as I think that there are many ways of explaining the bases for the binary differences.

The authors say that previous work suggests that the amygdala plays a role in selecting a learning rate, but aren't the references claiming that learning actually occurs in the amygdala? This would make a slightly different prediction from the claim here – which should make different predictions for figure 8 – namely that lesioned animals should fail to learn in the first place (eg. phase 1). Once again, for this reason, it would be useful to see model predictions in a richer format, alongside the relevant behavioral data if available. Furthermore, the authors should clarify why they believe that lesioning an area responsible for selecting volatility should lead the brain to think volatility is always zero. In many other domains, lesions can lead to higher reliance on a global prior, which I think would yield a different result.

I have a number of points of confusion regarding figure 9. First, why is the learning rate in control condition so high even in the stochastic condition? Optimal learning in this case is the average across outcomes... so if the model is providing a good approximation to this, it should be doing something similar, right? If the learning rate is 0.7, the model is making most of its errors by being overly switchy -- does this match monkey behavior? It strikes me as unlikely to be the case, but if the data are available it would certainly be a useful prediction to test, and much more specific than the overall error rates. Also, why do initialization of volatility and unpredictability differ so much across experiments (ie. figure 8 and figure 9)?

The anxiety predictions from the model suggest that “anxious” individuals should have higher learning rates across conditions – but this is not what was observed in Browning 2015. As I recall, anxious individuals in the empirical study used average learning rates similar to less anxious ones, and only differed in the degree to which they modulated learning across conditions. It would be useful if the authors could clarify, and if it exists, address this discrepancy. Once again, as with the other studies above, rich data exist for the empirical study that really nail down the dynamics of the learning behavior in anxious and non-anxious individuals – distilling this data to a single greater than or less than seems to throw out a lot of potentially useful information. As a minor point – the text for this section says:

“model shows insensitivity to volatility manipulation, but that is actually because volatility is misestimated nearer ceiling due to underestimation of volatility.” presumably the last volatility is a typo.

Finally, the modeling methods could be better elaborated. In particular I was unclear on how exactly particle filtering was implemented.

Minor concerns:

The authors say: “Note that while considered together, these two lines of studies separately demonstrate the two types of effects on learning rates we stress, neither of these lines of work has manipulated unpredictability alongside volatility” – but I think that Lee, Gold and Kable, 2020, does just this.

I have not seen “unpredictability” used before to mean measurement noise... but it does not seem very specific – as volatility also decreases outcome predictability. I would advocate usage of one of the existing terms, as there is already no shortage of such terms in the literature (observation noise, measurement noise, irreducible uncertainty).

Graphical model should show a second timestep so that it is apparent how autocorrelation emerges from volatility but not unpredictability.

Is figure 2 necessary? Since the model is just an extension of the idea in Behrens 2007 it seems like a mention that it can capture these basic effects should be sufficient. If the authors feel it is necessary, it would be good for it to match the actual experimental design, which had more

deterministic reward structure in the volatile task phase.

Why does the learning rate start out low in figure 4 simulations? A Kalman filter with a single drift and observation noise should start out with a learning rate near 1 that decays across trials. I don't see why this model should be different (or why starting with low learning would be useful).

Page 8: "The amygdala is known to be critical for associative learning"

The authors should clarify what they mean here – presumably the amygdala is not responsible for all forms of associative learning (many of which occur in organisms that don't have an amygdala).

Page 19: "though the accompanying models did not address our main question of how subjects estimate the noise hyperparameters"

Nassar 2010 did include models that inferred noise hyperparameters and showed a similar interaction to that which is focused on here.

Reviewer #2 (Remarks to the Author):

Unpredictability vs. volatility and the control of learning

Piray and Daw

In this paper the authors describe an hierarchical model that estimates two causes of outcome variability (unpredictability and volatility). They describe the underlying logic of the model and describe a series of simulations in which the predictions of the model are broadly compared to previously reported results.

The paper is well written and generally clear. The question of learning in the face of different sources of variability is topical with relatively less focus in the literature on what the authors term unpredictability.

The basic concepts explained in the paper are similar to those describe in the recent Pulcu and Browning (2019) paper, although the nomenclature is a little different and the current paper focuses more on simulations of previous results and provides detail on their particular model.

I had a couple of thoughts on what might improve the paper:

1. I think it would be useful to be a little more critical of the predictions generated by the model. The authors list quite a broad range of different studies and uncritically describe their model as capturing important details of the results. It would be useful to know a bit about the results the model is not able to capture. As an example, the authors suggest that their model can explain the results from studies of patients with anxiety if they force it to assume a low unpredictability. In their simulated results, the model uses a generally high learning rate (as it overestimates volatility). The authors describe this as being similar to their own previous results. However, this pattern of generally increased learning rates wasn't found in the Browning et al. 2015 study (Lrs were lower in the volatile condition), or in other work (there has not been a general finding that anxious individuals have high LRs). As it stands the authors report many different simulations briefly. I would suggest cutting the number of simulations down and being a little more critical of the ones they do report.
2. It would be useful to have a brief section with some concrete predictions of novel results arising from their model—for example, what sort of situation might promote the misattribution of volatility and unpredictability and what sort of data should we expect in these situations?

Reviewer #3 (Remarks to the Author):

Piray P and Daw ND, Unpredictability vs. volatility and the control of learning

The authors present a learning model in which they distinguish two types of stochasticity, unpredictability and volatility. They point out that volatility has been considered in several studies, but

unpredictability has received less attention. This distinction is important because the two forms of stochasticity push learning rates in opposite directions. Volatility should lead to an increase in learning rates, and unpredictability should lead to a decrease. They develop the model using a Kalman filter framework.

This paper presents an important idea. The paper is clearly written. The figures illustrate the concepts well. And the example datasets to which they fit their data are also appropriate and clear. I found Fig. 7 to be particularly important. Perhaps it would be worth moving Fig. 7 closer to the front of the results? The authors mention this result several times before the figure which shows it.

I have only minor comments.

Comments.

1. I was curious about the use of multiplicative noise models for the temporal evolution of volatility and unpredictability. Why not additive? I may have missed this, but perhaps a comment in the methods to clarify this choice. Does this help with the dissociation?

2. Reference to Figures 4 and 5 here should, I think be to Figures 5 and 6.

“Note the subtle difference between the experiments of Figures 4 and 5.”

3. In Fig. 7, it would be worth showing the healthy unpredictability and volatility, or referencing Fig. 4 for comparison.

4. I believe the volatility at the end of this sentence should say unpredictability, “...because volatility is misestimated nearer ceiling due to underestimation of volatility.”

5. These sentences have typos:

Notably, this type of finding cannot be explained by models like which learn only about volatility.

For both lesion models, lesioning does not merely abolish the corresponding effects on learning rate, but reverses it.

In particular, these of effects and a number of others are well explained by the unpredictability lesion model...

Reviewer #1 (Remarks to the Author):

Piray and Daw develop a particle filter-based approximate inference model for learning in environments with unknown “volatility” and “unpredictability”. They apply the model to explain seemingly incongruent results from conditioning in rodents, human behavior across multiple tasks, changes in behavior that emerge after amygdala lesions, and individual differences in learning that occur with anxiety.

This paper is impressive in that its scope. It attempts to capture data from three species in a wide range of task designs. In my opinion, a primary advantage of models is that they provide a way to generalize across the task specifics – and I think this paper makes an impressive contribution in that direction. However, I have a large number of concerns about the paper, many of which are probably consequences of the strategy taken by the authors to model many things, but not model anything in detail. My hope is that the authors are able to address these concerns, as I think if adequately revised the paper could provide a very nice contribution to the field.

Major concerns:

The authors motivate the novelty of their work based on a new model where volatility and unpredictability are learned simultaneously from experience. I would argue that they are not the first to do this – for example, Wilson 2010 provides an algorithm for optimal inference in environments with changepoints and unknown variance – that maps qualitatively onto the constructs explored here. Nor are the authors the first to explore the implications for the interaction between volatility and unpredictability estimation either theoretically (Yu & Dayan 2005) or in relation to individual differences in behavior (eg. noise estimation model from Nassar 2010). The implications for conditioning have also been described previously to some extent (Courville & Daw 2006) – and the implications for anxiety have also been discussed (Pulcu & Browning 2019). What has never been done previously, that is done in this manuscript, is to systematically apply one model to such a broad array of task findings across experimental designs and species. I think that is really valuable, and in my opinion it is what makes this work appropriate for a broad readership. But I think reframing could be necessary to clarify exactly what this work provides beyond the work listed above.

We thank the reviewer for their positive assessment of our work and for helpful suggestions, which helped us improve the manuscript substantially.

First, we have reworked different parts of the manuscript so as to make contributions of this work clearer. We agree with the reviewer that our main contribution is not the mechanics of simultaneously tracking both volatility/hazard and unpredictability/observation noise per se. This is, in practice, a challenging computational problem (and one where we are still hard-pressed to find quantitative testing and demonstration of successful solution in the literature), but it is in any case conceptually straightforward and not our main focus.

While we are thankful to the reviewer for appreciation of the broad scope of our paper, we also don't think that our contribution is limited to that. In the revision, we have tried to expose and focus on a single, novel theme that connects our contributions. Most of our varied applications to neuroscience, psychology, and psychiatry are consequences of a simple but widely applicable new insight, which we identify by raising unpredictability to the same status as volatility and focusing attention on their joint inference. *In particular, we show that estimating both types of hyperparameter simultaneously leads to issues of explaining away between them, with implications that are evident in healthy human and animal behavior, neurological and psychiatric disorders, and complicate the interpretation*

of much previous research. These issues only become clear when we put all the pieces together in one place and consider both aspects of hyperparameter inference simultaneously, and are not raised in the prior work.

With this in mind, we can revisit some of the previous work mentioned.

- 1) Obviously our work is most directly descended from the line on hierarchical Gaussian filters (from Behrens through Mathys and our VKF), which has pretty much universally focused on volatility estimation alone. For this reason, we motivate our direct contribution as building on this work by extending tracking to the second hyperparameter. But we try to make clear at several points that in so doing we are bringing these studies into contact with the research on changepoint tasks (from Nassar, Wilson and others), which we see as important to our argument because it demonstrates the contribution of tracking changing levels of unpredictability, balanced against volatility (i.e. hazard).

However – in contrast to the Behrens/Mathys research which is focused on volatility estimation – our overall read is that the changepoint studies have mostly not emphasized hyperparameter learning per se, vs. the within-block dynamics of first-level learning rates in this type of problem, with at least some hyperparameters taken as given or fit blockwise. Even where it has considered hyperparameters, this work has not to our knowledge considered the issues arising from simultaneous estimation/tracking of both.

Wilson et al. 2010 (Neural Computation) focus on estimating an unknown and changing hazard rate, which we take as conceptually related to volatility. We appreciate the reviewer’s calling our attention to the fact that this algorithm, while tracking a changing hazard, can also marginalize an unknown (but *fixed*) observation noise. We now cite this point. We could find no comment in that paper as to whether this is actually exercised in the example simulations, nor any overt exploration of this capability, but we agree this is a technical solution to a problem related to the one we address, of jointly estimating both parameters when both are potentially also changing.

However, when the implications of this model are taken into the empirical domain (in the line of studies starting with Nassar et al., 2010) the model is referenced but then replaced with a set of simplified algorithms in which only the noise is now inferred (eg Eq 22 of Nassar et al., 2010) conditioned upon a hazard that is taken as a fit free parameter (eg text following Eqs 14 and 25) – i.e. fixed and known from the perspective of the model’s inference. This simplification again skirts the main issue we pursue in the current study. As the reviewer points out, the fit hazard is viewed as characterizing an individual difference, and the study considers issues that arise from how this parameter (again, viewed inside the model as fixed and given) affects the model’s internal, dynamic noise estimation. We agree (and now mention) that this dependence has some resonance with the issues we explore, but we also think it’s fair to say that this literature just hasn’t addressed the points we make about the consequences of interdependent joint inference of hyperparameters and the phenomenon of explaining away.

- 2) We also do not think that Yu and Dayan (2005) solve, or even raise, the problems that we expose here. There are a few different variants of this model, but as we discuss, in the most general one (NIPS 2003), they are focused on distinguishing two types of change dynamics from one another (discrete changepoints from the combination of both Gaussian diffusion and observation noise in the baseline Kalman filter). It seems to us a misreading that later authors have sometimes identified their “expected and unexpected uncertainty” terminology with volatility and unpredictability in hierarchical Kalman filters. The models also do not address hyperparameter estimation at all (except insofar as, when a binomial observation model is used, the mean and unpredictability are the same quantity) – the emphasis, as with Nassar, is on changepoint detection.
- 3) Regarding some of the applications, while there have been applications of this type of model to Pavlovian conditioning, we think that the example of Courville, Daw and Tourtzky (2006) illustrates precisely what they lacked, because it presages our current account only with respect to Pearce/Hall, volatility, and surprise enhancement and latent inhibition of learning rates. However, at that time, we had missed the connection between unpredictability, Mackintosh, and effects like the partial reinforcement extinction effect (PREE). The model simply neglected unpredictability, in part because it used a binomial observation model. This had a specific impact on the attempt to account for the PREE (where we conceded we could not explain the results of Haselgrove et al. 2004 showing the effect is not driven by just outcome probability; we now model this experiment in Supplementary Fig 5) – but the broader and more important point is that this paper was typical of the literature attempting to connect Pavlovian associability models to statistical inference, in that it was not at the time conceptually clear which types of uncertainty and noise were at issue. Our current account of the mapping between the two hyperparameters, the two classes of effects in these types of studies, and the earlier psychological models is, to our knowledge, novel to the current paper, and again grows directly out of our consideration of the joint problem in light of the later modeling and experimental literature.
- 4) And finally, the short review by Pulcu and Browning (2019) presages our model in that they stress the importance of both types of uncertainty in the HGF models. We now mention this precedent sooner, more often, and more explicitly. However, our current paper presents a great deal of formal development, simulation and analysis to develop these ideas in much more detail – and especially, again, those about the interaction between the inference of the two variables and its implications, which they barely broach. Thus, while they mention the possibility that people might in principle confuse volatility and unpredictability, they identify none of the many implications of this possibility that our paper discusses, including in their discussion of anxiety.

On anxiety, they advocate the general view – which we share – that “misestimation of uncertainty is involved in the development of anxiety and depression.” However, neither of the two possible mechanisms they briefly sketch for this is the same as our proposals. The closer one is their recapitulation of the account from Browning et al. (2015), arguing that anxiety affects processes related to volatility estimation. This indeed is the starting point for our own discussion of anxiety. But our whole argument is that the apparent volatility effects they identify are secondary to more fundamental effects on unpredictability, and that the broader pattern of effects is deeply tied up

with this interrelationship. We develop this idea (and many more nuances and details, moreso in the current revision) connecting those results also to our own empirical work (Piray et al., 2019) and that of Huang et al. (2017) among others (note that an explanation that only considers volatility cannot explain and even contradicts with Huang et al. (2017), Harle et al (2017) and Aylward et al. (2019).)

They also present a second idea about anxiety that has little relationship with our work. Following their recent empirical work (Pulcu and Browning, 2017, elife), Pulcu and Browning stress that misestimation of uncertainty might be particularly related to its impact on symptoms of anxiety and depression “by skewing the processing of affective stimuli in a manner that favours negative over positive events” (Pulcu and Browning, 2019). We believe that this is a very promising hypothesis, but it is orthogonal to what we propose here.

My second major concern is that while the breadth of modeling applied here is commendable, the depth of explanation for each condition leaves something to be desired. Most of the major concerns below are different specific cases of this issue.

We are grateful to the reviewer for helpful suggestions. In light of the reviewer’s comments, we have done several new simulation analyses (Figs 5-6) and extended our previous ones (Supplementary Figs 2-8; Supplementary Table 1), which we believe deepens each problem that we consider substantially.

For example, figure 3 models a task with rich behavioral dynamics that have been extremely well characterized... but the only thing that is modeled is the difference in average learning rate under high and low noise conditions. While this is a feature of human behavior in that task, it is a minor one, with the more impressive features being the scaling of learning rate as a function of error magnitude and noise, and the scaling of learning rate with the number of trials since the last change. If the authors wish to make the claim that the discrete transitions in that task can be well modeled by the drifting process in their model it seems imperative to show that the model captures these core features of behavior.

Thanks for the comment. We have addressed these issues with additional simulations (discussed below). However, in light of both reviewers’ advice regarding focus and your suggestion to omit or downplay Figure 2 (the Behrens simulation which was the companion to this one), we have also moved this figure and the simulation of the Behrens task to supplement. We hope this helps to make the revised manuscript clearer about the role these data and their simulation play in our argument, because the aspects of it we emphasize and downplay again reflect what are the novel aspects of our work.

First, as we now say explicitly, we are not advocating our model as a superior overall account of the changepoint experiments, compared to models specialized for that task. To the contrary, we fully expect that a model, such as Nassar’s, which incorporates the underlying generative process of the task as it was instructed to subjects, should be able to outperform our model for this specific task.

We instead bring up these data because they are the best available demonstration of the sensitivity of human choice behavior to the outcome noise – importantly with the opposite effect on learning rate as has been emphasized for hazard/volatility. This is precisely the feature that we argue the HGF line of studies has neglected both experimentally and theoretically, and which our model incorporates. We agree that this blockwise adjustment

was not emphasized in Nassar’s original papers, even though it is prominent (e.g Nassar et al., 2010, Figure 2B). But we do not agree it is “minor,” particularly in the context of its relevance to the novel issues in our current study and to the adjacent literature on volatility.

Conversely, the pattern of learning rates around changepoints, while clearly of great interest and importance, is pretty well studied in both the HGF and changepoint lines of models, and we aren’t intending to add anything new about that. We don’t intend a strong commitment to the distributional form of the change (diffusion vs jump), since our issues of focus (the evidence that people must infer both change and noise hyperparameters, and the consequences of this) play out analogously in either formulation. Similarly, although again this is not a new insight and not our focus, we think the basic point about learning rate increasing at a changepoint due to inferred change (whether mediated by a local change in inferred volatility or an inferred jump) is also pretty similar, in that there are parametric regimes (i.e., fast volatility learning, appropriate to a situation when subjects are instructed about or experience jumps) in which the volatility model mimics the changepoint models in this respect. Indeed (as we now say in the paper) we understand the Wilson/Nassar approximate inference models (which approximate exact changepoint identification with a delta rule, i.e. the same form as the Kalman filter and its relatives) as a clear demonstration that seemingly different generative assumptions ultimately give rise to similar learning dynamics here. Overall, in modeling (empirical) discrete changepoints with inference derived from a Gaussian diffusion process, we aren’t trying to make a new or provocative claim here, just being faithful to a consistent feature of the preceding line of HGF models (back to Behrens 2007), which was arguably infelicitous but seems inconsequential for our current purposes.

With all that said, we have conducted further simulation analyses and have revised the related figure (now Supplementary Fig 2) to show that the model captures additional aspects of behavior in the Nassar’s task, as suggested by the reviewer. In particular, we have shown that the model can reproduce two aspects of empirical data in that task: 1) the learning rate increases following jumps and reduces over time; 2) the learning rate scales positively with the magnitude of error.

In the preceding line of studies, unpredictability was not manipulated. (Indeed, it was not even independently manipulable because rewards were binary, and the variance of binomial outcomes is determined only by the mean.) However, analogous effects of unpredictability have been seen in another line of studies (McGuire et al., 2014; Nassar et al., 2010, 2012, 2016). In these studies, Nassar and colleagues studied learning rates in task in which subjects had to predict a value, from observations in which the true value was corrupted, blockwise, by different levels of additive Gaussian noise (i.e., unpredictability) and occasionally “jumping” with a constant hazard rate, analogous to volatility. The main feature of these results relevant to the current model is that these studies have shown that participants’ learning rate decreases with increases in the noise level (see also (Lee et al., 2020)). This effect cannot be explained by models that only consider volatility, and in fact, those models make opposite predictions because they take increased noise as evidence of volatility increase. The current model, however, produces the same blockwise effect as humans: because it correctly infers the change in unpredictability, its learning rate is lower, on average, for higher levels of noise (Supplementary Figure 2). Although we do not intend the current model as a detailed account of how people solve this class of

tasks (which is based on a somewhat different generative dynamics), the model can also reproduce other more fine-grained aspects of human behavior in this task, particularly increases in learning rate following switches and scaling of learning rate with the magnitude of error (Supplementary Figure 2).

Supplementary Fig 2. The model reduces the learning rate in unpredictable environments. a) The prediction task by Nassar and colleagues, in which the participant makes a new prediction of outcome on every trial. Outcomes are generated based on a true reward rate, which undergoes occasional jumps, plus small or large amount of noise (i.e. true unpredictability). b) Behavior of the model in this task: Increases in the noise level is analogous to increases in unpredictability, which decreases the learning rate. The model also explains aspects of empirical data (Nassar et al., 2010) that are quite independent of unpredictability and are more closely related to jumps (c-d). c) Learning rate increases following switches in the task for both types of noise, although this effect is stronger for smaller noise level. d) The model learning rate also increases by increase in absolute error magnitude (divided by the value of true noise). Errorbars reflect standard error of the mean over 100 simulations.

For all simulations, but particularly the conditioning simulations, I found it very difficult to understand exactly what is being shown to the model (ie. how many trials per phase, how were values assigned to each condition). This makes interpretation of the model "behavior" difficult. But from what I can see, I suspect that this model diverges pretty systematically from the rodent behavior in that it uses learning rates that are extremely high. I know the data from several of the cited conditioning experiments quite well, and none that I know of show anything that looks like one shot learning. Acquisition and extinction curves are slow -- and based on the learning rates presented I would guess that this model vastly underestimates the number of trials that would be necessary to acquire/extinguish conditioning behavior. Claims could be strengthened by showing that the model predicts some aspects of behavior beyond the binary higher/lower comparison -- as I think that there are many ways of explaining the bases for the binary differences.

Thanks for these comments. Before the specifics on the conditioning experiments, a few general comments on the intention of the simulations, and how we have adjusted them in light of the reviewer's comments. In general, given the broad scope (and length) of the paper, our primary goal has been to expose a range of examples of how our main issue of interest (the interplay between volatility and unpredictability inference) plays out in different settings. We have, in turn, endeavored to extract and summarize evidence relevant to this main focus in each case, and unpack it in terms of the underlying volatility and unpredictability estimates, in a way that facilitates comparing across the different studies and appreciating the connecting theme.

However, we take seriously the reviewer's point that this strategy may stylize the results a bit too much, and obscure the way the issues play out in (real or simulated) data. In light of this, we have attempted to preserve the basic flow and approach of the manuscript, but accompanied each of these summary simulations with more disaggregated figures (usually learning curves or timeseries) in supplement. We hope this also helps to clarify the details of the simulations, which we have also tried to address with more detailed Methods simulation details throughout. Also, we take the point that each of these individual domains has many details, and our goal is more to trace our theme's applications through them; we now acknowledge that there may be other ways of explaining individual effects.

As for conditioning, we are grateful to the reviewer for pointing out that learning in rodent (appetitive) conditioning is usually slow. Of course, the net learning rate is parameter-dependent, but the overall pattern of effects we stress is relatively invariant. We have revised the simulations related to conditioning tasks to better capture this aspect of the data. For example, the current results entail lower learning rates across both conditions of Hall and Pearce experiment as well as those of partial reinforcement effect (Figure 6). Furthermore, we have also plotted the empirical data for both conditioning experiments (Supplementary Fig 4 and 5) along with model simulations in the current revision.

Figure 4. The model explains puzzling issues in Pavlovian learning. a-d) Pearce and Hall's conditioned suppression experiment. The design of experiment (Hall and Pearce, 1982), in which they found that the omission group show higher speed of learning than the control group (a). b) Median learning rate over the first trial of the retraining. The learning rate is larger for the omission group due to increases of volatility (c), while unpredictability is similar for both groups (d). The model explains partial reinforcement extinction effects (e-h). e) The partial reinforcement experiment consists of a partial condition in which a light cue is followed by reward on 50% of trials and a full condition in which the cue is always followed by the reward. f) Learning rate over the first trial of retraining has been plotted. Similar to empirical data, the model predicts that the learning rate is larger in the full condition, because partial reinforcements have relatively small effects on volatility (g), but it considerably increases unpredictability (h). Errorbars reflect standard error of the mean over 100 simulations and are, for some parameters, too small to be visible. See Supplementary Figures 4 and 5 for empirical data and corresponding response probability by the model.

And in the supplementary:

Supplementary Fig 4. Suppression ratio reported by Hall and Pearce (a) and the median response probability by the model in the retraining phase (b). The omission group shows faster decrease, consistent with empirical data. Suppression ratio has been defined as

the ratio of response in the 90 seconds window following presentation of the cue divided by its sum with the response rate in the preceding window of 90 seconds. Data in a adapted from Hall and Pearce (1982).

Supplementary Fig 5. Empirical data for the partial reinforcement effect experiment (a) and the median response probability by the model in the retraining phase (b). Empirical data of the retraining phase were reported by Haselgrove et al. (2004), in which the relative score has been calculated by subtracting the duration of magazine activity during the pre-CS period from duration of the magazine activity during the CS period. The average relative score across two sessions of extinction has been plotted.

The authors say that previous work suggests that the amygdala plays a role in selecting a learning rate, but aren't the references claiming that learning actually occurs in the amygdala? This would make a slightly different prediction from the claim here – which should make different predictions for figure 8 – namely that lesioned animals should fail to learn in the first place (eg. phase 1). Once again, for this reason, it would be useful to see model predictions in a richer format, alongside the relevant behavioral data if available. Furthermore, the authors should clarify why they believe that lesioning an area responsible for selecting volatility should lead the brain to think volatility is always zero. In many other domains, lesions can lead to higher reliance on a global prior, which I think would yield a different result.

Thank you for this comment. We have carefully considered this comment but we believe that our original understanding of previous work regarding the role of amygdala in learning rate is consistent with the position of the references, some of which we have indeed co-authored. It is true that some authors (e.g. LeDoux) have envisioned amygdala as a plasticity site for CS-US association per se, but here we are addressing a distinct line of work (especially due to Holland and Gallagher) arguing that it plays a more modulatory role as part of a circuit for controlling learning rate in conditioning. We would like to make it clear that the hypothesis that amygdala plays a critical role in adjusting the learning rate is not original here, and has been suggested by different authors: 1) Holland and Gallagher, in a series of lesion studies, suggest that rodents' (central nucleus of) amygdala is critical for upwards adjustments of associability in the Pearce-Hall sense; 2) Roesch et al. suggest a similar role for amygdala in learning based on their recording data (see for example Roesch et al., 2012); 3) We published a human fMRI study (Li et al., 2011) demonstrating that BOLD activity in amygdala covaries with learning rate or associability – as opposed to associative strength – in a conditioning task, and this study has engendered several

followups; 4) Costa and Averbeck (Costa et al., 2016; Averbeck and Costa, 2017) propose a similar role for amygdala based on their lesion studies in monkeys.

Regarding the specific suggestion that primary acquisition might be affected, we have now reported data from initial training from Holland and Gallagher (1993, our Supplementary Table 1) along with the same data from our model simulations. Unfortunately, these are present only as summary measures in the original study, but the bottom line is that lesioned animals were able to learn as well as control animals (Holland and Gallagher, 1993, 1999) in phase 1, and our simulations show similar behavior. Note that error bars have not been reported by Holland and Gallagher for the first phase, but they report no significant differences between groups.

Group	Data	Model
Control-consistent	53.9	56.22
Control-shift	55.6	55.96
Lesioned-consistent	59.9	56.21
Lesioned-shift	52.2	56.22

Supplementary Table 1. Reported data (time spent in the food cup%) of phase 1 of the experiment by Holland and Gallagher (Figure 7) following presentation of the second cue. No significant difference was found between control and lesioned animals suggesting that lesioned animals were able to learn efficiently. The model shows the same behavior (percentage of food response is reported, which is calculated based on a softmax with the decision noise of 0.5).

Finally, regarding the point about the prior, the reviewer is correct in pointing out that a lesion disabling adaptive inference of some parameter could be modeled by replacing it with a fixed, global prior. What we have done is indeed equivalent to this, assuming that prior has a very small value. We think this is a reasonable choice – emphasizing the explaining-away nature of the inference where different types of inferred noise compete to account for the observed variance. A hypothetical variant, which perhaps the reviewer has in mind, could assume a more moderate (but still fixed) volatility level in this situation, such that the volatility level under lesion is too high for some tasks but too low for others. Specifically for amygdala lesion, we think this less likely in light of the larger pattern of Holland and Gallagher’s studies (which show specific impairment with volatility-driven increases but not stability-driven decreases in learning rate). But in any case our results do not depend on assuming that volatility is literally 0 in the lesioned case, but simply on having it fixed at a level lower than what is required by the experiments we model.

I have a number of points of confusion regarding figure 9. First, why is the learning rate in control condition so high even in the stochastic condition? Optimal learning in this case is the average across outcomes... so if the model is providing a good approximation to this, it should be doing something similar, right? If the learning rate is 0.7, the model is making most of its errors by being overly switchy -- does this match monkey behavior? It strikes me as unlikely to be the case, but if the data are available it would certainly be a useful prediction to test, and much more specific than the overall error rates. Also, why do initialization of volatility and unpredictability differ so much across experiments (ie. figure 8 and figure 9)?

Thanks for this comment.

There are several questions here. First, as with the other parts of the paper, we now unpack the summary performance measure not just with timeseries of the models' internal parameters, but with full learning curves in supplement (Supplementary Fig 7), which corresponds to this figure of Costa et al (part of their Figure 2B). (We have also simulated the more detailed probabilistic schedules that Costa et al. tested; supplementary Figure 8.) As you see, there is an overall match between simulations and empirical data.

And simulations by the model:

Second, regarding learning rates, what constitutes optimal learning depends on the agent's belief/knowledge about the process parameters. Of course, *if the observer is fully informed that process noise is zero*, then a running average is optimal. This is not the case for the agent given our generative model and the prior parameters we have simulated, though as can be seen in the figure, it is gradually figuring that out. We chose these parameters (which net out to a median learning rate in the stochastic condition for controls of 0.48, not 0.7), in part because this seemingly fast learning (and correspondingly slow hyperparameter adjustment) is in fact relatively well matched with net learning rates estimated from the fit of an RL model by Costa et al. They estimate separate learning rates for positive and negative feedback (a detail we omit); but in this figure (Figure 4B reproduced from Costa et al.), the values of learning rate for controls (black) from positive and negative feedback are about 0.68 and 0.3, respectively, bracketing our net LR.

Finally, regarding initialization of volatility and unpredictability, we have endeavored to choose initial values that reproduce empirical behavior relatively well, and didn't expect or attempt to enforce that these should necessarily be consistent between species and tasks. Having said this, in the current revision we have revised simulations related to amygdala lesion studies (Figure 7 related to Holland), so as the initial hyperparameters are the same for both Figure 7 and 8.

Finally, as a broader point, a number of the reviewer's comments point to the observation that these parameters (as in this case) imply that animals' prior beliefs are pretty far from informed ideal observers in some particular laboratory task. We think this is an interesting point, but not in our view an objection to our argument and not really related to the novel issues we are trying to emphasize here – indeed, it has been a longstanding point of study on its own in the application of statistical models to animal conditioning. For instance, it is the central puzzle motivating Kakade and Dayan's, 2002, introduction of Kalman filter models to critique Gallistel and Gibbon's "optimal" rate estimation model.

The anxiety predictions from the model suggest that "anxious" individuals should have higher learning rates across conditions – but this is not what was observed in Browning 2015. As I recall, anxious individuals in the empirical study used average learning rates similar to less anxious ones, and only differed in the degree to which they modulated learning across conditions. It would be useful if the authors could clarify, and if it exists, address this discrepancy. Once again, as with the other studies above, rich data exist for the empirical study that really nail down the dynamics of the learning behavior in anxious and non-anxious individuals – distilling this data to a single greater than or less than seems to throw out a lot of potentially useful information. As a minor point – the text for this section says: "model shows insensitivity to volatility manipulation, but that is actually because volatility is misestimated nearer ceiling due to underestimation of volatility." presumably the last volatility is a typo.

We are grateful for this point, which has helped us to clarify the way in which our interpretation of anxiety effects on learning is different from the account put forward by Browning et al. (2015) and Pulcu and Browning (2019).

We now clearly state that, unlike a purely volatility-based account, we do predict overall higher learning rates, but that these should be most visible in more stable conditions (where control participants exhibit lower learning rates and the comparison is better powered). As we discuss, we think this is reasonably consistent with Browning et al.'s statistical results (they estimate that learning rate in the stable block has a positive, though not significant, relationship with anxiety; $r(28)=0.26$; $P=0.16$). It is of course inappropriate to conclude that an effect is either present *or absent* given a nonsignificant result, but if

anything, this is more consistent with our hypothesis than the opposite (i.e. the learning rate in the stable block is not modulated by anxiety.)

More importantly, we now discuss that our view is also supported by positive evidence from additional studies (about which we have added more simulation and discussion) that are better suited than Browning's to investigate this matter, and which are consistent with the view that anxiety increases learning rate regardless of volatility manipulation. Notably, compared with Browning et al (2015), these studies have tested more subjects and have recruited them based on trait anxiety (therefore they compare two groups of low- and high- anxious individuals, with more substantial difference in trait anxiety). In the current version, we have conducted new simulations and have shown that our model simulates the main result of Huang et al (2017), who report that anxiety **does** modulate average learning rate, as well as win-stay/lose-shift behavior in a more powerful study (n=122) and with rigorous criteria for defining low- and high- anxiety groups.

One key prediction of our model, which differs from a volatility-specific account, is that the learning rate is generally higher in anxious people regardless of volatility manipulation or even in tasks that do not manipulate volatility. In fact, Browning et al. (2015) do not find evidence to support this prediction: they do not find a significant overall effect of anxiety on learning rate. Of course, it is important not to interpret null results as evidence in favor of the null hypothesis, since a failure to reject the null hypothesis may reflect insufficient power to detect a true effect. Indeed, in Browning's (2015) data, while the effect of anxiety on learning rate was not significant overall or in either volatility condition, the point estimate was largest ($r(28)=0.26$, $p=0.16$) in the stable condition, which is also the block in which the model predicts the effect should be statistically strongest (because baseline learning rates, absent any effect of anxiety, are lower).

Importantly, other, larger studies provide positive statistical support for the prediction of elevated learning rate with anxiety (Aylward et al., 2019; Harlé et al., 2017; Huang et al., 2017). Note that in delta-rule models, behavior under higher learning rates is closer to win-stay/lose-shift (since higher learning rates weight the most recent outcome more heavily, with full WLS – dependence only on the most recent outcome – equivalent to a learning rate of 1). Such a strategy has itself been linked to anxiety (Harlé et al., 2017; Huang et al., 2017). A notable observation was made in a large (n=122) study by Huang et al. (2017), who found anxious people show higher win-stay/lose-shift and this effect is driven by higher lose-shift. Figure 5gh shows results of simulating the proposed model in a task similar to Huang et al. (2017) (Supplementary Figure 6). The model shows the same pattern of behavior, with the additional modulation by win vs loss captured because any loss is seen as an evidence for volatility and that results in higher learning rate and a contingency switch. The effect is much less salient for win trials because prediction errors are relatively small in those trials, which substantially dampen any effect of learning rate. Across all trials, the anxious model shows higher learning rate, similar to what Huang et al (2017) found by fitting reinforcement learning models to choice data.

Figure 5. The unpredictability lesion model shows a pattern of learning deficits associated with anxiety. Behavior of the lesioned model as the model of anxiety, in which unpredictability is assumed to be small and constant, is shown along the control model. a-d) Behavior of the models in the switching task of Figure 2 is shown. An example of estimated reward by the models shows that the anxious model is more sensitive to noisy outcomes (a), which dramatically reduces sensitivity of the learning rate to volatility manipulation in this task (b). This is, however, primarily related to inability to make inference about unpredictability, which leads to misestimation of volatility (c-d). e-f) The model explains the data reported by Piray et al. (2019), in which the high (social) anxiety group did not benefit from stability as much as the low anxiety group (e). The model shows the same behavior (f). g-h) The model explains the data by Huang et al. (2017), in which the anxious group showed higher lose-shift behavior compared to the control group (g). The model shows the same behavior (g), which is due to higher learning rate in the anxious group (inset). Errorbars reflect standard error of the mean.

Finally, as for the other points about the details of behavior, please also note that, unfortunately, there is not much available about the dynamics of learning behavior in

Browning et al. study, beyond what we already cover, as the main focus of that study was on pupil dilatory data. However, there is an additional between-subjects correlational finding, and in an attempt to address the reviewer’s comment here we have now also conducted further simulation analysis and reproduced it (Figure 6).

Finally, the lesion model is an extreme case in which a hypothetical unpredictability module is completely eliminated. But this general approach can be extended to less extreme cases in which one module of the model (e.g. unpredictability) has some relative disadvantage in explaining noise. In terms of our model, this can be achieved by having higher update rate parameters for volatility relative to that of unpredictability. These are two main parameters of the model that one can use to explain individual differences across people. For example, the ratio of volatility to unpredictability update rate can be used to capture continuous individual variation in trait anxiety. In this case, the unpredictability lesion model of Figure 3b is an extreme case of this approach in which the unpredictability update rate is zero (thus the ratio of volatility to unpredictability is infinitely large). We have exploited this approach to simulate a result from Browning et al. (2015) concerning graded individual differences in anxiety’s effect on learning rate adjustment. In particular, they report (and the model captures; Figure 6) negative correlation between relative learning rate (stable minus volatility) and trait anxiety in the probabilistic switching task of Figure 5a.

Figure 6. The model explains effects of trait anxiety as a continuous index on learning. a) Data by Browning et al. (2015) show a significant negative correlation between relative log learning rate (stable minus volatile block.) b) The model shows a similar pattern. The inset shows the median rank correlation between trait anxiety and the relative learning rate over 100 simulations. Model trait anxiety is defined as the ratio of volatility to unpredictability update rates. The lesion model of anxiety (Figure 5) is a special case in which the unpredictability update rate is zero. Errorbars reflect standard error of the median.

Finally, the modeling methods could be better elaborated. In particular I was unclear on how exactly particle filtering was implemented.

Thank you for this final point. We agree and have endeavored to make methods, including details of implementation of the particle filter, clearer. We also endeavored to make the code (which was publicly available and cited in the manuscript) clearer.

For inference, we employed a Rao-Blackwellised Particle Filtering approach (Doucet et al., 2000), in which the inference about z_t and u_t were made by a particle filter (Doucet and Johansen, 2011) and, conditional on these, the inference over x_t was given by the Kalman filter (Eqs 1-4). The particle filter is a Monte Carlo sequential importance sampling method, which keeps track of a set of particles (i.e. samples). The algorithm performs three steps on each trial. First, in a prediction step, each particle is transitioned to the next step based on the generative process. Second, weights of each particle are updated based on the probability of observed outcome:

$$b_t^l \propto N(o_t | m_{t-1}^l, w_{t-1}^l + v_t^l + u_t^l)$$

where b_t^l is the weight of particle l on trial t , m_{t-1}^l and w_{t-1}^l are estimated mean and variance by the Kalman filter on the previous trial (Eqs. 1-4), and v_t^l and u_t^l are volatility and unpredictability samples (i.e. the inverse of z_t^l and y_t^l). In this step, particles were also resampled using the systematic resampling procedure if the ratio of effective to total particles fall below 0.5. In the third step, the Kalman filter (Eqs. 1-4) was used to update the mean and variance. In particular, for every particle, Eqs 1-4 were used to calculate α_t^l and update m_t^l and w_t^l . Learning rate and estimated reward rate on trial t was then defined as the weighted average of all particles, in which the weights were given by b_t^l . We have used particle filter routines implemented in MATLAB.

Minor concerns:

The authors say: “Note that while considered together, these two lines of studies separately demonstrate the two types of effects on learning rates we stress, neither of these lines of work has manipulated unpredictability alongside volatility” – but I think that Lee, Gold and Kable, 2020, does just this.

Thanks for this point. We have noted this paper more than once in the current version.

I have not seen “unpredictability” used before to mean measurement noise... but it does not seem very specific – as volatility also decreases outcome predictability. I would advocate usage of one of the existing terms, as there is already no shortage of such terms in the literature (observation noise, measurement noise, irreducible uncertainty).

Thanks for this suggestion. We have considered it carefully and are not sure how to proceed. We agree that “unpredictability” is less specific than would be ideal, and also that “process noise” vs “observation (or measurement) noise” is standard, and more precise terminology. Unfortunately, we are building directly on a literature in which the term “volatility” is already in use, so we feel we need a term that goes with it. The lack of parallelism between “volatility” and “observation noise” seems to imply incorrectly that one is a type of noise, but not the other. We also think that variations on “uncertainty” are incorrect, because in our view “uncertainty” should properly be reserved for the subjective posterior rather than (the true value of, or beliefs about the true value of) objective, generative stochasticity. For now, we have introduced our terminology with explicit connection to process and measurement noise, but have stuck with “unpredictability” thereafter. However, in light of this comment, we have also considered switching to

“stochasticity” – which is perhaps a little more specific. We’re open to further advice here – is stochasticity better? Are we missing another term?

Graphical model should show a second timestep so that it is apparent how autocorrelation emerges from volatility but not unpredictability.

Done!

Is figure 2 necessary? Since the model is just an extension of the idea in Behrens 2007 it seems like a mention that it can capture these basic effects should be sufficient. If the authors feel it is necessary, it would be good for it to match the actual experimental design, which had more deterministic reward structure in the volatile task phase.

Thanks for this suggestion. In the light of this comment, and also suggestions by other reviewers, we have decided to move this figure, along with its parallel figure regarding unpredictability (i.e. Nassar’s simulation), to supplementary.

Why does the learning rate start out low in figure 4 simulations? A Kalman filter with a single drift and observation noise should start out with a learning rate near 1 that decays across trials. I don’t see why this model should be different (or why starting with low learning would be useful).

Thanks for this point. We agree and have revised that simulation to have learning rates started from near 1. To be clear, in general, it is not the case that the Kalman filter always shows a declining learning rate – this depends on how much prior uncertainty it has about the state at time zero, relative to the levels of known process and observation noise that ensue thereafter. (A classic example where this would be useful is a rocket which blasts off from a fully known position, after which volatility contributes to state uncertainty and the resulting gain grows, rather than falls, to asymptote.) Previously, our model was in a similar situation to this because we had chosen a priori state variance to be 1 for simplicity, a choice which (given the scale of the other parameters) led to the situation of initially increasing gain in that figure. Although this does not affect the qualitative results with respect to the points being emphasized here (the blockwise inference), we agree it is a slightly odd choice in this type of experiment, and have changed the prior accordingly.

Page 8: “The amygdala is known to be critical for associative learning”

The authors should clarify what they mean here – presumably the amygdala is not responsible for all forms of associative learning (many of which occur in organisms that don’t have an amygdala).

Thanks for the comment. We have changed the wording of that sentence in the revision:

The amygdala plays an important role in associative learning (Averbeck and Costa, 2017; Phelps et al., 2014).

Page 19: “though the accompanying models did not address our main question of how subjects estimate the noise hyperparameters”

Nassar 2010 did include models that inferred noise hyperparameters and showed a similar interaction to that which is focused on here.

Thanks for the comment. We have revised the text accordingly.

Reviewer #2 (Remarks to the Author):

In this paper the authors describe an hierarchical model that estimates two causes of outcome variability (unpredictability and volatility). They describe the underlying logic of the model and describe a series of simulations in which the predictions of the model are broadly compared to previously reported results.

The paper is well written and generally clear. The question of learning in the face of different sources of variability is topical with relatively less focus in the literature on what the authors term unpredictability. The basic concepts explained in the paper are similar to those describe in the recent Pulcu and Browning (2019) paper, although the nomenclature is a little different and the current paper focuses more on simulations of previous results and provides detail on their particular model.

We are grateful for the reviewer's positive evaluation of our manuscript and for their helpful suggestions.

Regarding Pulcu and Browning (2019), we discuss the relationship between this important prior work and ours earlier and more clearly in the revision (and also in our response to the first reviewer, above). We very much accord with and now more expressly build upon their general perspective, but we would like to make it clear that we do not see that this review comprises the novelty of our work. That paper is a short review/opinion paper about possible associations between misestimation of uncertainty and anxiety. It contains no formal equations or systematic simulation of any data, and one figure of some type of model run but without any methods or code. As we now say, their review does emphasize some important themes that we develop in our more formal modeling study, especially the importance of inferring both of what we call volatility and unpredictability to healthy and disordered behavior.

As the current revision hopefully makes clearer, our most important novel claims in the current study all center around the observation that jointly inferring these two parameters leads (due to explaining away) to rich patterns of interaction between them, which have implications across several domains of healthy and disordered behavior, and also invite revisiting the interpretation of numerous previous results. Pulcu and Browning (2019) do mention, briefly, that people might misattribute or confuse these types of uncertainty, but do not develop the implications of this point. In particular, their suggestions about how these processes are actually implicated in anxiety do not develop or reflect this idea; as discussed at greater length below, this leads us to a novel and quite distinct interpretation of Browning et al.'s (2015) results compared to the one they recapitulate there. (They also include a second idea, about valence-specific processing, which is quite promising but orthogonal to the points in our current review, and not mutually exclusive.) The scope of the current work is also much wider than anxiety, including applications to neurological damage and to conditioning.

1. I think it would be useful to be a little more critical of the predictions generated by the model. The authors list quite a broad range of different studies and uncritically describe their model as capturing important details of the results. It would be useful to know a bit about the results the model is not able to capture. As an example, the authors suggest that their model can explain the results from studies of patients with anxiety if they force it to assume a low unpredictability. In their simulated results, the model uses a generally high learning rate (as it overestimates volatility). The authors describe this as being similar to their own previous results. However, this pattern of generally increased learning rates wasn't found in the Browning et al. 2015 study (Lrs were lower in the volatile condition), or in other work (there has not

been a general finding that anxious individuals have high LRs). As it stands the authors report many different simulations briefly. I would suggest cutting the number of simulations down and being a little more critical of the ones they do report.

Thanks for the comment, which we have addressed in a number of ways. First, we have emphasized a number of aspects in which the modeling is abstracted or simplified relative to details of the original studies (e.g., changepoints vs diffusion; asymmetric learning rates in the monkey studies; various limitations in anxiety). We have also, more generally, stated more clearly our strategy of exposing the common theme (explaining away between volatility and unpredictability) as it may arise or have implications in many different situations and settings, while conceding that these individual situations all involve many more details and may individually admit of different interpretations. But we have also deepened our account of these areas by adding several more simulations, elaborating previous ones in many different directions (Supplementary Figs 2-8; Supplementary Table 1). We have included several new simulations regarding the anxiety part, including of the Huang et al. (2017) study and also an additional result from the Browning et al. (2015) study.

With respect to the points about anxiety specifically, we are grateful to the reviewer (and see also related comments from Reviewer 1 and our response above) for pointing out this issue. We have addressed it much more systematically with more simulation and discussion. We believe this is a case which helps us to explain how our model offers a different interpretation of the Browning results, which we would defend as a better explanation, overall, of those results in the broader pattern of results in the literature.

It is true that Browning et al. (2015), conclude there is no relationship between overall learning rates and anxiety on the basis of statistically null results. Our account indeed predicts a positive relationship here, but also that it will be most easily detectable in the stable regime (where learning rates are otherwise lower) and not in the volatile block (where they more compressed toward high values even without anxiety). On our view, this is reasonably consistent with the underlying (nonsignificant) cell-wise estimates in Browning's data: they estimate a nonsignificant but distinctly positive association in the stable block ($r=0.26$, $p=0.16$) and one much nearer to zero ($r=-0.1$, $p=0.6$) in the volatile block. (Perhaps it is this latter estimate to which the reviewer is referring when he or she says that "LRs were lower in the volatile condition"? Clearly, LRs were significantly higher in the volatile condition compared with the stable one across all subjects.)

Clearly it is inappropriate to draw firm conclusions either way from null results, a point we make more clearly in the present revision. But we also include new discussion and simulations of other studies (which we would argue are better designed and powered for addressing this specific question) that we think support our view, and are pertinent to the reviewer's point about a more general finding. Indeed, **there are multiple other studies from various other labs that report higher learning rates in anxious individuals**, which we enumerate here: 1) In a large sample study ($n=122$), Huang et al (2017) have found anxious individuals show higher base learning rate (their Fig 4A; reproduced here for convenience) and higher win-stay/lose-shift. In the current revision, we have simulated these results too.

2) Aylward et al. (2019) have reported higher learning rate for learning from punishment in anxious individuals in a bandit task that did not modulate volatility at all. (Relatedly, we have now discussed the possibility that there might be a distinct, but not mutually exclusive, direction about the role of positive and negative valence on learning rate in anxious individuals.)

3) Harle et al (2017) have found that anxiety is positively correlated with the ability (i.e. model evidence) of a win-stay/lose-shift model to explain choice data in a bandit task, which is clearly consistent with our hypothesis that the anxiety increases learning rate substantially. Again, volatility has not been manipulated in this study.

We have added new simulations and revised the text to address these points:

These results have been interpreted in relation to the more general idea that intolerance of uncertainty is a key foundation of anxiety; accordingly, fully understanding them any require taking account of multiple sources of uncertainty (Pulcu and Browning, 2019), including both volatility and unpredictability. Nevertheless, the primary interpretation of these types of results has been that observed abnormalities are rooted in volatility estimation per se (Browning et al., 2015; Piray et al., 2019; Pulcu and Browning, 2019). Our current model suggests an alternative explanation: that the core underlying deficit is actually with unpredictability, and apparent disturbances in volatility processing are secondary to this, due to their interrelationship.

In particular, these effects and a number of others are well explained by the unpredictability lesion model of Figure 3b, i.e., by assuming that anxious people have a core deficit in estimating unpredictability, and instead treat it as small and constant. As shown in Figure 5b, this model shows insensitivity to volatility manipulation, but in the model that is actually because volatility is misestimated nearer ceiling due to underestimation of unpredictability. This, in turn, substantially dampens further adaptation of learning rate in blocks when volatility actually increases. The elevated learning rate across all blocks leads to hypersensitivity to noise, which prevents anxious individuals from benefitting from stability, as has been observed empirically (Piray et al., 2019). In particular, Piray et al. have studied learning in individuals with low- or high- in trait social anxiety using a switching probabilistic task (Supplementary Figure 6) in which each trial started with a social threatening cue (angry face image). It was found that anxious individuals perform particularly worse than controls in stable trials, whereas their performance is generally matched with controls in volatile trials (Figure 5e). The model shows similar behavior (Figure 5f).

One key prediction of our model, which differs from a volatility-specific account, is that the learning rate is generally higher in anxious people regardless of volatility manipulation or even in tasks that do not manipulate volatility. In fact, Browning et al. (2015) do not find evidence to support this prediction: they do not find a significant overall effect of anxiety on learning rate. Of course, it is important not to interpret null results as evidence in favor of the null hypothesis, since a failure to reject the null hypothesis may reflect insufficient power to detect a true effect. Indeed, in Browning's (2015) data, while the effect of anxiety on learning rate was not significant overall or in either volatility condition, the point estimate was largest ($r(28)=0.26$, $p=0.16$) in the stable condition, which is also the block in which the model predicts the effect should be statistically strongest (because baseline learning rates, absent any effect of anxiety, are lower).

Importantly, other, larger studies provide positive statistical support for the prediction of elevated learning rate with anxiety (Aylward et al., 2019; Harlé et al., 2017; Huang et al., 2017). Note that in delta-rule models, behavior under higher learning rates is closer to win-stay/lose-shift (since higher learning rates weight the most recent outcome more heavily, with full WLS – dependence only on the most recent outcome – equivalent to a learning rate of 1). Such a strategy has itself been linked to anxiety (Harlé et al., 2017; Huang et al., 2017). A notable observation was made in a large ($n=122$) study by Huang et al. (2017), who found anxious people show higher win-stay/lose-shift and this effect is driven by higher lose-shift. Figure 5gh shows results of simulating the proposed model in a task similar to Huang et al. (2017) (Supplementary Figure 6). The model shows the same pattern of behavior, with the additional modulation by win vs loss captured because any loss is seen as an evidence for volatility and that results in higher learning rate and a contingency switch. The effect is much less salient for win trials because prediction errors are relatively small in those trials, which substantially dampen any effect of learning rate. Across all trials, the anxious model shows higher learning rate, similar to what Huang et al (2017) found by fitting reinforcement learning models to choice data.

Figure 5. The unpredictability lesion model shows a pattern of learning deficits associated with anxiety. Behavior of the lesioned model as the model of anxiety, in which unpredictability is assumed to be small and constant, is shown along the control model. a-d) Behavior of the models in the switching task of Figure 2 is shown. An example of estimated reward by the models shows that the anxious model is more sensitive to noisy outcomes (a), which dramatically reduces sensitivity of the learning rate to volatility manipulation in this task (b). This is, however, primarily related to inability to make inference about unpredictability, which leads to misestimation of volatility (c-d). e-f) The model explains the data reported by Piray et al. (2019), in which the high (social) anxiety group did not benefit from stability as much as the low anxiety group (e). The model shows the same behavior (f). g-h) The model explains the data by Huang et al. (2017), in which the anxious group showed higher lose-shift behavior compared to the control group (g). The model shows the same behavior (g), which is due to higher learning rate in the anxious group (inset). Errorbars reflect standard error of the mean.

2. It would be useful to have a brief section with some concrete predictions of novel results arising from their model—for example, what sort of situation might promote the misattribution of volatility and unpredictability and what sort of data should we expect in these situations?

Thanks for this suggestion. We actually suggest that the most pressing experimental issue raised by the model is primarily about simply generating the basic data needed to constrain such an account, i.e. manipulating both types of noise factorially as in the hypothetical experiment of Figure 2. With such data in hand, it might be possible to refine and compare different, detailed trial-wise accounts of the dynamics of attribution. Although it is likely possible to “fool” the model and cause healthy subjects to misattribute (and perhaps this is what the reviewer has in mind), our ideas about misattribution at the current time mainly focus on the hypothetical patterns of pathological or individual differences. Our key novel prediction (Figure 3) is that in cases of manipulations or pathology where effects have been seen on volatility or learning rate, more careful examination will reveal a reciprocal pattern. This is the extreme case of misattribution. In light of this comment, we also considered and simulated less extreme cases of misattribution, and patterns of more graded individual differences that might arise. We have now discussed this issue more extensively, particularly in the context of anxiety. Furthermore, we have done further simulation analyses and showed that our framework is able to model trait anxiety (as a continuous measure) and reproduce the main behavioral result of Browning et al. (2015):

Finally, the lesion model is an extreme case in which a hypothetical unpredictability module is completely eliminated. But this general approach can be extended to less extreme cases in which one module of the model (e.g. unpredictability) has some relative disadvantage in explaining noise. In terms of our model, this can be achieved by having higher update rate parameters for volatility relative to that of unpredictability. These are two main parameters of the model that one can use to explain individual differences across people. For example, the ratio of volatility to unpredictability update rate can be used to capture continuous individual variation in trait anxiety. In this case, the unpredictability lesion model of Figure 3b is an extreme case of this approach in which the unpredictability update rate is zero (thus the ratio of volatility to unpredictability is infinitely large). We have exploited this approach to simulate a result from Browning et al. (2015) concerning graded individual differences in anxiety’s effect on learning rate adjustment. In particular, they report (and the model captures; Figure 6) negative correlation between relative learning rate (stable minus volatility) and trait anxiety in the probabilistic switching task of Figure 5a.

Figure 6. The model explains effects of trait anxiety as a continuous index on learning. a) Data by Browning et al. (2015) show a significant negative correlation between relative log learning rate (stable minus volatile block.) b) The model shows a similar pattern. The inset shows the median rank correlation between trait anxiety and the relative learning rate over 100 simulations. Model trait anxiety is defined as the ratio of volatility to unpredictability update rates. The lesion model of anxiety (Figure 5) is a special case in which the unpredictability update rate is zero. Errorbars reflect standard error of the median.

Reviewer #3 (Remarks to the Author):

The authors present a learning model in which they distinguish two types of stochasticity, unpredictability and volatility. They point out that volatility has been considered in several studies, but unpredictability has received less attention. This distinction is important because the two forms of stochasticity push learning rates in opposite directions. Volatility should lead to an increase in learning rates, and unpredictability should lead to a decrease. They develop the model using a Kalman filter framework.

This paper presents an important idea. The paper is clearly written. The figures illustrate the concepts well. And the example datasets to which they fit their data are also appropriate and clear. I found Fig. 7 to be particularly important. Perhaps it would be worth moving Fig. 7 closer to the front of the results? The authors mention this result several times before the figure which shows it.

I have only minor comments.

We are very grateful for the reviewer's positive evaluation of our manuscript, recognition of its novelty and for the insightful and helpful suggestions. We agree with the reviewer that it is helpful to present the lesioned figure earlier in the results. We have moved it and present it as Figure 3 of the current version, which is right after the simulation results of the healthy model.

1. I was curious about the use of multiplicative noise models for the temporal evolution of volatility and unpredictability. Why not additive? I may have missed this, but perhaps a comment in the methods to clarify this choice. Does this help with the dissociation?

Thanks for this great point. We are not really committed to a specific generative process (e.g. a multiplicative one). The reason for choosing that generative process was that it is the same process that we used in our recent work (Piray and Daw, 2020). In turn, we had adopted it there because it eliminates a nonlinearity (if variances follow an additive Gaussian walk, they can go negative) and also because it facilitates exact solution of one of the maximizations in a variational approximation, which is not used in the current particle filter implementation. We have now revised Figure 1 and Methods to make this clear. We have also conducted further simulation analyses to show that the model also works well with an alternative generative process (Gaussian) as long as the same particle filter has been used (Supplementary Fig 3).

2. Reference to Figures 4 and 5 here should, I think be to Figures 5 and 6.

“Note the subtle difference between the experiments of Figures 4 and 5.”

Done!

3. In Fig. 7, it would be worth showing the healthy unpredictability and volatility, or referencing Fig. 4 for comparison.

Done!

4. I believe the volatility at the end of this sentence should say unpredictability, “...because volatility is misestimated nearer ceiling due to underestimation of volatility.”

Thanks for pointing this typo out.

5. These sentences have typos:

Notably, this type of finding cannot be explained by models like which learn only about volatility.

For both lesion models, lesioning does not merely abolish the corresponding effects on learning rate, but reverses it.

In particular, these of effects and a number of others are well explained by the unpredictability lesion model...

Thanks for pointing these typos out. And thanks again for helpful suggestions.

REVIEWER COMMENTS

Reviewer #1 (Remarks to the Author):

The manuscript is considerably improved and to me it is clear that it makes an important contribution. I have just one remaining concern, related to the terminology (unpredictability)”

Personally, I don't see any issue with the term observation noise. Avoiding problematic misconceptions by introducing new terminology obscures relationships to existing literature and sets a bad precedent. It seems that a more useful approach would be to use the established terminology and to provide clear definitions that correct these misconceptions – thereby making life easier for the next person who publishes a paper in this field

Reviewer #2 (Remarks to the Author):

The authors have provided a detailed response to my initial review. They do include some caveats to their analyses in the discussion now, which are welcome.

The new section on anxiety is interesting, the proposal that there is a generally raised I_r in response to misestimated unpredictability is specific enough to be new I think. This would seem consistent with the previous finding that anxious individuals acquire fear conditioning more rapidly, although it seems difficult to square this with the more consistent finding that anxiety is associated with reduced extinction of fear.

If the authors think this isn't a problem for their proposal, it might be useful to add a line explaining this. If it doesn't fit, then this is worth acknowledging (as it is probably the most commonly reported finding in anxiety conditioning studies).

Reviewer #3 (Remarks to the Author):

The authors have addressed my concerns. I have no fourth comments.

Reviewer #1 (Remarks to the Author):

The manuscript is considerably improved and to me it is clear that it makes an important contribution. I have just one remaining concern, related to the terminology (unpredictability)”

Personally, I don't see any issue with the term observation noise. Avoiding problematic misconceptions by introducing new terminology obscures relationships to existing literature and sets a bad precedent. It seems that a more useful approach would be to use the established terminology and to provide clear definitions that correct these misconceptions – thereby making life easier for the next person who publishes a paper in this field.

We are thankful to the reviewer for recognizing the contributions of our work. Regarding the terminology, we take the reviewer's' points on this issue seriously and have considered them carefully, but we still feel that we need a term for unpredictability that goes in parallel with the already established term volatility, and it is misleading here to call only one of them “noise”. More particularly, if anything, the correct technical term for what we called “unpredictability” is the “variance of the observation noise” and not the “observation noise” itself (or, as we originally called this, the observation noise parameter, simply because the observation noise has only one parameter). This is similar to volatility, where the term is used primarily to refer to the variance governing the process noise (akin to the hazard *rate*), not the process noise itself. Given the importance of this term in the current work, we feel that we also need a one-word term here. In the light of this comment, however, we feel that a slightly better term for the latent variable that we called unpredictability would be stochasticity, which we have used in this final version.

Reviewer #2 (Remarks to the Author):

The authors have provided a detailed response to my initial review. They do include some caveats to their analyses in the discussion now, which are welcome.

The new section on anxiety is interesting, the proposal that there is a generally raised I_r in response to misestimated unpredictability is specific enough to be new I think. This would seem consistent with the previous finding that anxious individuals acquire fear conditioning more rapidly, although it seems difficult to square this with the more consistent finding that anxiety is associated with reduced extinction of fear.

If the authors think this isn't a problem for their proposal, it might be useful to add a line explaining this. If it doesn't fit, then this is worth acknowledging (as it is probably the most commonly reported finding in anxiety conditioning studies).

We are thankful to the reviewer for their positive evaluation of the revised manuscript, and for these important points about fear conditioning. Our interpretation would be that the seeming disconnect between acquisition and extinction reflects the fact that unlike acquisition, extinction is dominated by other processes (notably state splitting / contextual inference) rather than associative (un) learning per se. Combining this type of model with inferred volatility and unpredictability is an important (but substantial) piece of future work; it is clear there should be substantial effects but it's not at present obvious (to us) how they would ultimately shake out in terms of faster or slower extinction in different circumstances. We now briefly mention this point in discussion:

This hypothesis is also consistent with the observation that acquisition of fear conditioning tends to be enhanced in anxious individuals (Duits et al., 2015; Lissek et al., 2005). Finally, although a simple increase in learning rate seems harder to reconcile with generally slower extinction of Pavlovian fear learning in anxiety (Duits et al., 2015), this probably reflects the well-known fact that extinction is not simply unlearning of the original association, but instead is dominated by additional processes (Bouton, 2004; Redish et al., 2007). This includes in particular statistical inference about latent contexts (Gershman et al., 2010), which is likely to be affected by both stochasticity and volatility in ways that should be explored in future work.